# New Bounds for Kernel Sums via Fast Spherical Embeddings

**Tal Wagner** [1]

## Abstract

We study query time bounds for the fundamental problem of estimating the kernel mean $\frac{1}{|X|} \sum_{x \in X} \mathbf{k}(x, y)$ of a query $y$ in a finite dataset $X \subset \mathbb{R}^d$ up to a prescribed additive error $\varepsilon$. The best known bounds for the Gaussian kernel are $O(d/\varepsilon^2)$, $\widetilde{O}(d + 1/\varepsilon^4)$, and $\widetilde{O}(d + \Delta^2/\varepsilon^2)$, where $\Delta$ is the diameter of a region containing the points. We prove the new bound $\widetilde{O}(d + \varepsilon\Delta^2 + 1/\varepsilon^3)$, which improves over the previous ones in regimes with small error $\varepsilon$ and intermediate diameter $\Delta$.

At the center of our proof is a new fast spherical embedding theorem in the sense introduced by Bartal, Recht and Schulman (2011), which limits the embedded data diameter while preserving local Euclidean distances and avoiding "distance collapse" at larger scales. This fast embedding theorem may be of independent interest.

## 1. Introduction

Estimating the empirical kernel density of a point in a finite dataset is a long-studied problem in machine learning with widespread use. We study the following data structure formulation of the kernel density estimation (KDE) problem.

**Definition 1.1.** Let $\mathbf{k} : \mathbb{R}^d \times \mathbb{R}^d \to \mathbb{R}$ be a map that we call the kernel. A randomized $(\varepsilon, \delta)$-KDE data structure $\mathcal{D}_{\mathbf{k}, \varepsilon, \delta}$ is defined as follows. $\mathcal{D}_{\mathbf{k}, \varepsilon, \delta}$ is constructed once over a finite set $X \subset \mathbb{R}^d$. Given each fixed query $y \in \mathbb{R}^d$, the goal is for $\mathcal{D}_{\mathbf{k}, \varepsilon, \delta}$ to report a KDE estimate $\widetilde{E}(y)$ such that

$$\Pr\left[\left|\left|\frac{1}{|X|} \sum_{x \in X} \mathbf{k}(x, y) - \widetilde{E}(y)\right| < \varepsilon\right] > 1 - \delta,\right.$$

where the probability is over the construction randomness of $\mathcal{D}_{\mathbf{k}, \varepsilon, \delta}$. Our main interest is in minimizing the query time.

This problem is well-studied and has given rise to classical methods like the Fast Gauss Transform (Greengard & Strain,

1991) and Random Fourier Features (Rahimi & Recht, 2007) (see Sections 2.1 and 5 for a review of prior work).

We focus mostly on the Gaussian kernel $\mathbf{k}(x, y) = \exp(-\|x - y\|_2^2/\sigma^2)$, where $\sigma > 0$ is the bandwidth parameter, and on the high-dimensional regime where exponential dependence on $d$ is prohibitive. We assume that $\Delta \in [0, \infty)$ is an upper bound on the diameter of a region $\mathcal{W} \subset \mathbb{R}^d$ that contains all data and query points used in KDE. Since the kernel is shift-invariant, we may assume w.l.o.g. that $\mathcal{W}$ is the $d$-dimensional origin-centered ball of diameter $\Delta$, which we denote by $\mathbb{B}^d(\Delta)$. We call $\Delta_\sigma := \Delta/\sigma$ the *effective diameter*. In most contexts we will work with $\sigma = 1$ (and thus $\Delta = \Delta_\sigma$) in order to ease notation. This does not limit generality as we can simply scale all points by $\sigma^{-1}$.

In all that follows we use the notation $\widetilde{O}(\cdot)$ to suppress constants and polylogarithmic factors in $d$, $\varepsilon^{-1}$, $\Delta_\sigma$ and $\delta^{-1}$ (or $\eta^{-1}$, as we will use $\eta$ for the failure probability in some contexts to avoid overloading notation).

In this setting, the currently best bounds known for Gaussian KDE query time as defined in Definition 1.1 are:

- $O(d/\varepsilon^2)$, by random Fourier features (RFF);

- $\widetilde{O}(d + 1/\varepsilon^4)$, by RFF composed over the Fast Johnson-Lindenstrauss Transform (FJLT), a fast Euclidean dimension reduction result due to Ailon & Chazelle (2009), as proposed and analyzed by Backurs et al. (2024);

- $\widetilde{O}(d + \Delta_\sigma^2/\varepsilon^2)$, by the Fastfood method (Le et al., 2013).

The bounds are incomparable and depend on the interplay between the parameters. The first two bounds entail no limitation on the diameter, while Fastfood improves over them if the (effective) diameter is sufficiently small relative to $d$ and $\varepsilon^{-1}$. The problem addressed in this work is to improve over those bounds.

### 1.1. Main Results

Our main result is a new bound for Gaussian KDE.

**Theorem 1.2.** *There is a Gaussian KDE data structure as in Definition 1.1 with query time* $\widetilde{O}(d + \varepsilon\Delta_\sigma^2 + 1/\varepsilon^3)$.

Like Fastfood, and unlike RFF and RFF+FJLT, the bound in Theorem 1.2 depends on $\Delta_\sigma$. However, the dependence is more favorable than in Fastfood, and it improves rather

---

[1]Tel Aviv University, Israel. Correspondence to: Tal Wagner <talwag@tauex.tau.ac.il>.

*Proceedings of the 43rd International Conference on Machine Learning*, Seoul, South Korea. PMLR 306, 2026. Copyright 2026 by the author(s).

*Table 1.* Summary of KDE query time bounds satisfying Definition 1.1 for the Gaussian kernel $\mathbf{k}(x, y) = \exp(-\|x - y\|_2^2/\sigma^2)$. $\Delta \in (0, \infty]$ is an upper bound on the diameter of a region that contains all points and $\Delta_\sigma = \Delta/\sigma$ is the effective diameter. The bounds here treat $\delta$ as a small constant and assume that $\varepsilon \ll 1$ and that the dimension $d$ is high.

| METHOD | REFERENCE | QUERY TIME | REGIME WHERE BEST |
|---|---|---|---|
| RFF | Rahimi & Recht (2007) | $O(d/\varepsilon^2)$ | $d \lesssim \varepsilon^{-2}$ and $\Delta_\sigma \gtrsim \sqrt{d}\varepsilon^{-1.5}$ |
| FJLT+RFF | Backurs et al. (2024) | $\widetilde{O}(d + 1/\varepsilon^4)$ | $d \gtrsim \varepsilon^{-2}$ and $\Delta_\sigma \gtrsim \varepsilon^{-2.5}$ |
| Fastfood | Le et al. (2013) | $\widetilde{O}(d + \Delta_\sigma^2/\varepsilon^2)$ | $\Delta_\sigma \lesssim \min\{\sqrt{d}, \varepsilon^{-0.5}\}$ |
| Ours | Theorem 1.2 | $\widetilde{O}(d + \varepsilon\Delta_\sigma^2 + 1/\varepsilon^3)$ | $\varepsilon^{-0.5} \lesssim \Delta_\sigma \lesssim \min\{\sqrt{d}\varepsilon^{-1.5}, \varepsilon^{-2.5}\}$ |

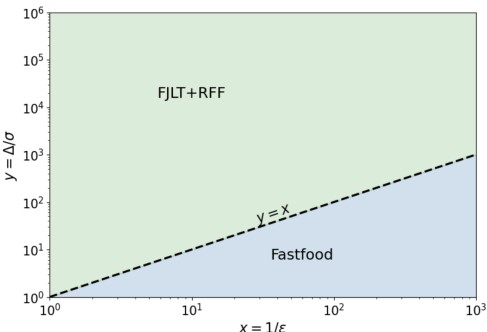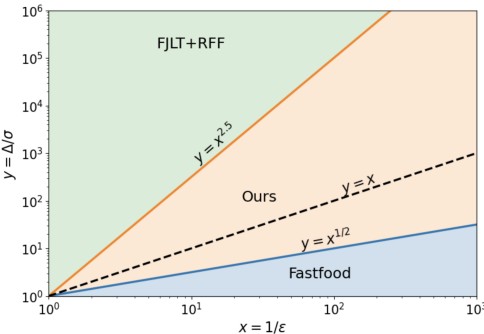

*Figure 1.* Best regime per method in Table 1, before (left) and after (right) ours, for Gaussian KDE in the high-dimensional case $d \gtrsim 1/\varepsilon^2$ (in this regime, RFF is always subsumed by at least FJLT+RFF). The plots depict the trade-off between the inverse-error $1/\varepsilon$ on the x-axis and the effective diameter $\Delta_\sigma = \Delta/\sigma$ on the y-axis. The axes are in log scale.

than degrades as $\varepsilon$ becomes smaller. The comparison of all bounds and regimes appears in Table 1 and Figure 1.

The main ingredient in our proof of Theorem 1.2 is a new fast spherical embedding theorem. It is essentially an FJLT-analog for the embedding of Bartal et al. (2011, Lemma 6) which they termed a "randomized Nash device".

**Theorem 1.3.** *Let* $\varepsilon, \eta \in (0, 1)$ *and* $\Lambda > 0$. *Let* $m = \widetilde{O}(d + \Lambda^2 + \varepsilon^{-2})$. *There is a randomized map* $\Phi : \mathbb{R}^d \to \mathbb{S}^m$, *computable in time* $\widetilde{O}(m)$, *such that for each fixed pair* $x, y \in \mathbb{R}^d$ *the following holds with probability* $1 - \eta$:

1. $\|\Phi(x) - \Phi(y)\|_2^2 \leq (1 + \varepsilon)\|x - y\|_2^2$.

2. $\|x - y\|_2^2 \leq \varepsilon \Rightarrow \|\Phi(x) - \Phi(y)\|_2^2 \geq (1 - \varepsilon)\|x - y\|_2^2$.

3. $\varepsilon < \|x - y\|_2^2 \leq \Lambda^2 \Rightarrow \|\Phi(x) - \Phi(y)\|_2^2 \geq \Omega(\varepsilon)$.

Bartal et al. (2011)'s embedding is essentially similar, except that it does not have $\Lambda$ and its running time is $O(d/\varepsilon^2)$. They introduced their embedding not for KDE but for a different set of applications; see Section 6 for a discussion. Theorem 1.3 may similarly find applications beyond KDE.

To summarize our main contributions:

- We prove a new time complexity bound for KDE queries, improving in some regimes over cornerstone methods

like RFF, FJLT and Fastfood.

- We achieve this by a new fast spherical embedding theorem, an FJLT-like analog to a result of Bartal et al. (2011).

- At the technical level, our proof introduces a new analysis of randomized Hadamard transforms and of the Fastfood method, based on a fourth chaos analysis (Section 3.3).

*Remark.* To justify our focus on the query time of KDE, we discuss its interaction with other complexity measures relevant to Definition 1.1, namely the construction time and space usage. All of the methods in Table 1 are based on linear features. This is discussed in detail in Section 2, but in short, they construct a feature map $x \mapsto f(x)$ into $m$ dimensions. Let $\mathcal{T}_f$ denote the time to apply $f$ to a point. The KDE data structure is constructed by computing and storing $F(X) := \frac{1}{|X|} \sum_{x \in X} f(x)$ in time $O(|X| \cdot \mathcal{T}_f)$. A query is answered by returning $F(X)^T f(y)$ in time $\mathcal{T}_Q := O(\mathcal{T}_f + m)$. Hence, the space usage is $m \leq \mathcal{T}_Q$ and the construction time is $O(|X| \cdot \mathcal{T}_Q)$. Therefore, $\mathcal{T}_Q$ captures the overall cost of the KDE data structure. In other words, there is no hidden lack of efficiency in the construction time or space usage of the data structure. Furthermore, it is known that as a preprocessing step, the size of $X$ can be reduced to $O(\log(1/\delta)/\varepsilon^2)$, either by standard random sampling or by a coreset (see Lemma A.1). This is why the query time does not need to depend on $|X|$ at all (even logarithmically).

## 1.2. Extensions

We show two extensions of our results to other settings of interest. The first is another standard and well-studied family of kernels: the inverse multi-quadratic (IMQ) kernels, also known as rational quadratic kernels, defined as $\mathbf{k}_\beta^{\mathrm{IMQ}}(x,y) = (1 + \|x - y\|_2^2/\sigma^2)^{-\beta}$ for $\beta > 0$.

**Theorem 1.4.** *Let $\beta_0 > 0$ be an arbitrarily small constant. For all $\beta \geq \beta_0$, there is a KDE data structure as in Definition 1.1 for $\mathbf{k}_\beta^{\mathrm{IMQ}}(x,y)$ with query time $\widetilde{O}(d + \varepsilon(\beta\Delta_\sigma)^2 + 1/\varepsilon^3)$. The $\widetilde{O}(\cdot)$ notation here also hides a $\log^{O(1)}(\beta)$ term.*

Theorem 1.4 is obtained by combining Theorem 1.2 with a function approximation result due to Beylkin & Monzón (2010), following an approach proposed by Backurs et al. (2024). The proof is in Appendix B.

The second extension is to differential privacy (DP). The RFF and FJLT+RFF bounds in Table 1 are known to extend to $\varepsilon_{\mathrm{DP}}$-DP KDE in the function release model, under the condition that $|X| \geq O(\frac{\log(1/\delta)}{\varepsilon^2 \varepsilon_{\mathrm{DP}}})$ (Wagner et al., 2023; Backurs et al., 2024). Our method as well as Fastfood necessitate a different treatment because of the probabilistic dependence across their coordinates, which is created by the randomized Hadamard transform they are based on. We show that they too extend to DP KDE under essentially the same condition. See Appendix C for details and proof.

**Theorem 1.5.** *There is an $\varepsilon_{\mathrm{DP}}$-DP KDE function release mechanism (see Definition C.1) with the same construction and query times and memory usage as Theorem 1.2, under the condition that $|X| \geq \widetilde{O}(1/(\varepsilon^2 \varepsilon_{\mathrm{DP}}))$.*

## 2. Technical Overview and Related Work

This section provides a high-level outline of our method in the context of prior work. We focus on the Gaussian kernel with $\sigma = 1$ and treat $\delta$ as a fixed small constant for simplicity. We denote by $\psi : \mathbb{R}^d \to \mathbb{R}^{2d}$ (for any dimension $d$) the mapping of a vector $x = (x_i)$ to the sine/cosine pairs of its entries, $\psi(x) = \oplus_i (\sin(x_i), \cos(x_i))$. We also denote $d_\varepsilon = \Theta(1/\varepsilon^2)$ for brevity and assume that $d \gg d_\varepsilon$.

### 2.1. Background and Prior Work

A basic approach in KDE data structures as defined in Definition 1.1 is through representing the kernel with approximate linear features. At construction time the data structure selects a (possibly randomized) map $f : \mathbb{R}^d \to \mathbb{R}^{d'}$ and computes $F(X) = \frac{1}{|X|} \sum_{x \in X} f(x)$. At query time it returns the estimate $F(X)^T f(y)$. The query time is thus $O(d')$ plus the time it takes to evaluate $f$ on $y$. In the low-dimensional case, where running times may depend exponentially on $d$, the Fast Gauss Transform (Greengard & Strain, 1991) is a classical method that takes this form.

In the high-dimensional case, a cornerstone technique is the RFF method of Rahimi & Recht (2007). Their feature map is $\psi(Wx)$, where $W \in \mathbb{R}^{d_\varepsilon \times d}$ has i.i.d. Gaussian entries. The query time is thus $O(d d_\varepsilon) = O(d/\varepsilon^2)$.

Randomized Hadamard Transforms (RHTs) are a powerful way to speed up methods based on unstructured random matrices (Sarlos, 2006; Ailon & Chazelle, 2009; Tropp, 2011; Drineas et al., 2011; Boutsidis & Gittens, 2013; Lu et al., 2013; Ailon & Liberty, 2013; Andoni et al., 2015; Yu et al., 2016; Choromanski et al., 2017). The normalized Hadamard matrix of order $2^\ell \times 2^\ell$ is defined inductively as

$$H_{2^0} := \begin{bmatrix} 1 \end{bmatrix} \quad , \quad H_{2^\ell} := \frac{1}{\sqrt{2}} \begin{bmatrix} H_{2^{\ell-1}} & H_{2^{\ell-1}} \\ H_{2^{\ell-1}} & -H_{2^{\ell-1}} \end{bmatrix}.$$

We will denote it by $H$ and let its order be inferred from context. An RHT is a random matrix of the form $HD$ where $D$ is a diagonal matrix with i.i.d. entries (whose distribution may vary by context, e.g., Gaussian or Rademacher). On the one hand, the matrix-vector product $HDx$ can be computed in time $O(d \log d)$ with the Walsh-Hadamard transform. On the other hand, $HD$ often exhibits similar properties to a matrix with i.i.d. Gaussian entries, rendering the RHT a useful proxy for it. Ailon & Chazelle (2009) used RHTs in the seminal Fast Johnson-Lindenstrauss Transform (FJLT) to speed up the classical JL Euclidean dimension reduction theorem (Johnson & Lindenstrauss, 1984), reducing its running time from $O(d d_\varepsilon)$ to $\widetilde{O}(d + d_\varepsilon)$

RHTs have been applied to kernel approximation in various ways. Backurs et al. (2024) analyzed FJLT as preprocessing for RFF, obtaining random features of the form $x \mapsto \psi(WSHDx)$, where $S \in \mathbb{R}^{d_\varepsilon \times d}$ is a row-subsampling matrix and $W \in \mathbb{R}^{d_\varepsilon \times d_\varepsilon}$ is a full Gaussian matrix. Note that while the RHT acts as Euclidean dimension reduction (FJLT) from $d$ to $d_\varepsilon$, a full Gaussian matrix $W$ is still required in their analysis for kernel approximation with RFF. The overall query time is $\widetilde{O}(d + d_\varepsilon^2) = \widetilde{O}(d + 1/\varepsilon^4)$.

Le et al. (2013) introduced the RHT-based Fastfood method. We define it formally in Section 3.1, but broadly speaking, it uses two iterated RHTs sequentially, taking the random feature form $x \mapsto \psi(SHD_2HD_1x)$.[1] We remark that iterated RHTs (often more than two) have been shown to be advantageous in various contexts (Yu et al., 2016; Andoni et al., 2015; Choromanski et al., 2017; 2018). Here $S \in \mathbb{R}^{d' \times d}$ is a row-subsampling matrix into $d' = O(\Delta^2/\varepsilon^2)$ dimensions, assuming that the points are known to be in a region of bounded diameter $\Delta$, as is often the case. The running time is $\widetilde{O}(d + \Delta^2/\varepsilon^2)$, which improves over the previous approaches if $\Delta$ is sufficiently small.

Cherapanamjeri & Nelson (2022) proved that concatenating $t = \widetilde{O}(\Delta^2/\varepsilon^2)$ feature maps $\{\psi(HD_jx)\}_{j=1}^t$ yields a fea-

---

[1] Le et al. (2013) discuss several Fastfood variants; see Remark 3.1. We use a simplified form which suffices for our purposes.

ture map that approximately preserves the Gaussian kernel simultaneously for *all* pairs of points in a bounded region of diameter $\Delta$. While the query time $O(dt) = \widetilde{O}(d\Delta^2/\varepsilon^2)$ is considerably increased, their "for all pairs" guarantee is stronger than the "for each pair" guarantee achieved by the methods in Table 1.

## 2.2. Our Method

Our starting point is Fastfood, which improves the dependence on $d, \varepsilon$ at the expense of introducing the dependence on diameter in the term $\Delta^2/\varepsilon^2$. A natural strategy for improvement is to add a preprocessing step that controls the diameter. At the same time, it must not distort point distances in a way that would distort the kernel estimate.

A useful observation is that while we may need to accurately preserve "small" distances, we can afford to only loosely preserve "large" ones. If $x, y \in \mathbb{R}^d$ are at distance $\|x - y\| \geq \sqrt{\log(1/\varepsilon)}$, then $e^{-\|x-y\|_2^2} \leq \varepsilon$. Thus, to ensure it is approximated it up to $\pm\varepsilon$, we do not need to preserve $\|x - y\|_2^2$; we need only ensure that it remains larger than $\sqrt{\log(1/\varepsilon)}$ and does not "collapse" to be any smaller.

Thus, we need preprocessing that *(i)* controls the diameter, *(ii)* preserves "small" distances, and *(iii)* keeps "large" distances from collapsing. Fortunately, this type of embedding has been introduced by Bartal et al. (2011) for a different set of applications. Their result embeds points in the unit sphere while preserving distances smaller than $\sqrt{\varepsilon}$ up to $(1 \pm \varepsilon)$ and preventing larger distances from collapsing below $\Omega(\sqrt{\epsilon})$. By scaling the "small" distance threshold, their embedding can be shown to preserve the Gaussian KDE.

Unfortunately, applying Bartal et al. (2011)'s embedding costs the same $O(d/\varepsilon^2)$ running time as RFF. In fact, their embedding is identical to RFF: it also has the form $x \mapsto \psi(Wx)$ where $W$ is a full Gaussian matrix of order $d_\varepsilon \times d$. Therefore, it cannot be used to improve the running time of KDE queries beyond what RFF already gives us.

This discussion suggests the strategy of proving a result similar to Bartal et al. (2011) but with a shorter running time. This is precisely the main ingredient in our approach, Theorem 1.3. Since their embedding has the form $\psi(Wx)$, a natural candidate is to replace the Gaussian matrix $W$ with an RHT-based proxy. We show that (a variant of) the Fastfood transform, $\psi(HD_2HD_1x)$, forms an embedding into the unit sphere with the properties we need.

Having proven the spherical embedding in Theorem 1.3, we can apply it as a preprocessing step for KDE. To adjust the "small" distance threshold from $\sqrt{\varepsilon}$ to our requisite $\sqrt{\log(1/\varepsilon)}$, we scale the points by $s = \Theta(\sqrt{\varepsilon/\log(1/\varepsilon)})$ on the way into Theorem 1.3 and "un-scale" by $s^{-1}$ on the way out. This means that the original diameter $\Delta$ yields the scaled diameter $\Lambda = s\Delta = \widetilde{O}(\sqrt{\varepsilon}\Delta)$ in Theorem 1.3, and

that the embedded un-scaled points lie on a sphere of radius $s^{-1}$, hence their new diameter is $\widehat{\Delta} := 2s^{-1} = \widetilde{O}(1/\sqrt{\varepsilon})$. We then apply Fastfood again on the embedded points, this time as intended and analyzed in Le et al. (2013) for the purpose of approximating the KDE (their result is cited here as Theorem 4.1). Put together, our feature map ultimately takes the form of two iterated Fastfood transforms:

$$f_{\text{ours}}(x) = \psi(HD_4HD_3 \cdot s^{-1}\psi(HD_2HD_1(sx))).$$

The inner Fastfood takes time $m = \widetilde{O}(d + \Lambda^2 + \varepsilon^{-2}) = \widetilde{O}(d + \varepsilon\Delta^2 + \varepsilon^{-2})$ by Theorem 1.3, and the outer Fastfood takes time $\widetilde{O}(m + \widehat{\Delta}^2/\varepsilon^2)$ by Le et al. (2013). (The difference is in their different output dimensions.) Since $\widehat{\Delta} = \widetilde{O}(1/\sqrt{\varepsilon})$, the time to apply $f_{\text{ours}}$ is $\widetilde{O}(d + \varepsilon\Delta^2 + \varepsilon^{-3})$, and this dominates the query time in Theorem 1.2.

The two Fastfood invocations in our method play different roles – one as a fast spherical embedding per Theorem 1.3, the other as a fast KDE approximation per Theorem 4.1. They have different proofs that require divergent techniques. The Fastfood analysis in Le et al. (2013) is based on concentration of Lipschitz functions of Gaussians. While sufficient for proving the kernel approximation guarantee in Theorem 4.1, this approach is insufficient for proving the spherical embedding guarantees in Theorem 1.3, particularly the bounded contraction property in item 2, since lower-bounding the trigonometric functions in Fastfood requires controlling a 4th order term in the random diagonal entries of the RHT. Our analysis overcomes this through a new analysis of Fastfood that applies a Wiener chaos decomposition (Wiener, 1938) and controls the 4th chaos term. This is done in the contraction proof in Section 3.3.

## 3. Fast Spherical Embedding

In this section we prove Theorem 1.3. Omitted proofs appear in Appendix A.

### 3.1. Preliminaries

**Assumptions.** We start with assumptions that will simplify the analysis without limiting generality. We set $m = \widetilde{O}(d + \Lambda^2 + \varepsilon^{-2})$ where the $\widetilde{O}(\cdot)$ notation hides a $\log^{O(1)}(d\Lambda/(\epsilon\eta))$ term of sufficiently high degree and $m$ is rounded up to a power of 2. We also assume w.l.o.g. that $d = m$ by zero-padding the input points up to dimension $m$. The target sphere will be $\mathbb{S}^{2m-1}$. It suffices to prove the theorem assuming $\varepsilon \leq \varepsilon_0$ where $\varepsilon_0$ is a sufficiently small constant. Finally, we will prove the theorem with $O(\varepsilon), O(\eta)$ instead of $\varepsilon, \eta$. This does not change the theorem as hidden constants can be rescaled.

**Fastfood.** We use a variant of the Fastfood transform due to (Le et al., 2013). Let $H$ be the normalized Hadamard matrix of order $m \times m$. It has entries in $\pm 1/\sqrt{m}$, satisfies $H^T H =$

$I$ (where $I$ is the order-$m$ unit matrix), and for every $x \in \mathbb{R}^m$ the matrix-vector product $Hx$ can be computed in time $O(m \log m)$ with the Walsh-Hadamard transform.

Let $G = \text{diag}(g) \in \mathbb{R}^m$ be a diagonal matrix with random i.i.d. Gaussian entries $g_j \sim N(0,1)$. Let $B \in \mathbb{R}^m$ be a random diagonal sign matrix where each diagonal entry is uniform in $\{\pm 1\}$. The Fastfood matrix $V \in \mathbb{R}^{m \times m}$ is

$$V = \sqrt{m} \cdot HGHB. \qquad (1)$$

*Remark* 3.1. Le et al. (2013) discuss several variants of Fastfood. The full form is $\sqrt{m} \cdot \Sigma HG\Pi HB$ where $\Sigma$ is a random scaling matrix and $\Pi$ is a random permutation matrix. We use the form (1) since it is the minimal one and it suffices for proving Theorem 1.3. Nevertheless, our proof also works with the full Fastfood variant. The matrices $\Sigma, \Pi$ are not necessary for the proof nor they interfere with it.

The Fastfood map $\Phi : \mathbb{R}^m \to \mathbb{R}^{2m}$ for which we prove Theorem 1.3 is defined, for every $j = 1, \ldots, m$, as

$$\Phi(x)_{2j-2} = \tfrac{1}{\sqrt{m}} \cos((V_x)_j),$$
$$\Phi(x)_{2j-1} = \tfrac{1}{\sqrt{m}} \sin((V_x)_j).$$

Note that while $\Phi$ is a randomized map, the following properties hold deterministically:

- $\|\Phi(x)\|_2^2 = 1$ for every $x \in \mathbb{R}^m$.
- Since $V$ is the product of diagonal and Hadamard matrices, the matrix-vector product $Vx$ for every $x \in \mathbb{R}^m$ can be computed in time $O(m \log m)$.

**Notation and basic properties.** To prove items 1–3 in Theorem 1.3 we fix a pair $x, y \in \mathbb{R}^m$ and denote $z = x - y$. We furthermore denote

$$u := HBz = HB(x - y).$$

Since both $H$ and $B$ are orthogonal matrices we have

$$\|u\|_2 = \|z\|_2. \qquad (2)$$

We now cite some key lemmas from prior work. Let

$$L_z := \|z\|_2 \sqrt{\frac{2 \log(2m/\eta)}{m}} \qquad (3)$$

Ailon & Chazelle (2009) proved the following flattening lemma for the randomized Hadamard transform.

**Lemma 3.2** (Ailon & Chazelle (2009)). *For each $z \in \mathbb{R}^m$,*

$$\Pr_B \left[ \|HBz\|_\infty \le L_z \right] > 1 - \eta.$$

We condition on this event throughout, losing an additive $\eta$ in the total probability. We thus have

$$\|u\|_\infty \le L_z. \qquad (4)$$

This is the only property we need of $B$. Henceforth we consider $B$ as fixed and satisfying Equation (4). The only source of randomness in the remainder of the proof is the diagonal Gaussian matrix $G = \text{diag}(g)$.

We will also need the following properties of Fastfood, proven in Le et al. (2013).

**Lemma 3.3** (Le et al. (2013)). *For every $j = 1, \ldots, m$, $(Vz)_j$ is distributed like $N(0, \|z\|_2^2)$.*

Recall that a function $f : \mathbb{R}^m \to \mathbb{R}$ is called $L$-Lipschitz if $|f(g') - f(g'')| \le L \cdot \|g' - g''\|_2$ for all $g', g'' \in \mathbb{R}^m$.

**Lemma 3.4** (Le et al. (2013)). *Let $f_z : \mathbb{R}^m \to \mathbb{R}$ be defined as $f_z(g) = \frac{1}{m} \sum_{j=1}^m \cos((Vz)_j)$. Then $f_z$ is $L_z$-Lipschitz.*

### 3.2. Distance Expansion

In this section we prove item 1 in Theorem 1.3, that $\Phi$ does not expand squared distances by more than $(1 + \varepsilon)$ with high probability. We start with following fact which follows from standard trigonometric identities (see Appendix A).

**Lemma 3.5.** *Deterministically for every supported $V$,*

$$\|\Phi(x) - \Phi(y)\|_2^2 = \frac{2}{m} \sum_{j=1}^m \left(1 - \cos((Vz)_j)\right).$$

By a Taylor expansion, $1 - \cos(\theta) \le \frac{1}{2}\theta^2$ for all $\theta$. Therefore, by Lemma 3.5,

$$\|\Phi(x) - \Phi(y)\|_2^2 \le \frac{1}{m} \sum_{j=1}^m ((Vz)_j)^2 =: Q(z). \qquad (5)$$

Observe that, since $H$ is orthogonal,

$$Q(z) = \frac{1}{m} \|Vz\|_2^2 = \|HGu\|_2^2 = \|Gu\|_2^2 = \sum_{j=1}^m g_j^2 u_j^2.$$

Therefore, since $\forall_j g_j \sim N(0,1)$ and by Equation (2),

$$\mathbb{E}[Q(z)] = \|u\|_2^2 = \|z\|_2^2.$$

Let $X_j = g_j^2 - 1$. Then $X_j$ is zero-centered and subexponential, and $\sum_{j=1}^m u_j^2 X_j = Q(z) - \mathbb{E}[Q(z)]$. By the subexponential Bernstein inequality (e.g., Vershynin (2018, Theorem 2.9.1), for a universal constant $c > 0$ and all $t \ge 0$,

$$\Pr \left[ \left| \sum_{i=1}^m u_j^2 X_j \right| > t \right] \le \exp \left( -c \cdot \min\{\frac{t^2}{\sum_j u_j^4}, \frac{t}{\|u\|_\infty^2}\} \right)$$

We take $t = \varepsilon \|z\|_2^2$. Using Equations (2) and (4), we have,

$$\sum_j u_j^4 \le \|u\|_\infty^2 \sum_j u_j^2 = \|u\|_\infty^2 \|u\|_2^2 \le L_z^2 \|z\|_2^2. \qquad (6)$$

Therefore,

$$\frac{t^2}{\sum_j u_j^4} \geq \frac{\varepsilon^2 \|z\|_2^4}{L_z^2 \|z\|_2^2} = \frac{\varepsilon^2 \|z\|_2^2}{L_z^2} \quad \text{and} \quad \frac{t}{\|u\|_\infty^2} \geq \frac{\varepsilon \|z\|_2^2}{L_z^2},$$

and the quantity on the left is smaller since $\varepsilon < 1$. Plugging this with Equation (3) into the Bernstein inequality,

$$\Pr\left[\left|Q(z) - \|z\|_2^2\right| \leq \varepsilon \|z\|_2^2\right] \geq 1 - \exp\left(-\frac{c\varepsilon^2 m}{2\log(m/\eta)}\right).$$

Recalling our setting of $m$, we get

$$\Pr\left[(1-\varepsilon)\|z\|_2^2 \leq Q(z) \leq (1+\varepsilon)\|z\|_2^2\right] \geq 1 - \eta. \quad (7)$$

Item 1 of Theorem 1.3 follows from Equations (5) and (7).

## 3.3. Distance Contraction

In this section we prove item 2 in Theorem 1.3, that $\Phi$ does not contract small squared distances by more than $(1-\varepsilon)$ with high probability, through a Wiener chaos analysis.

By a Taylor expansion, $1 - \cos(\theta) \geq \frac{1}{2}\theta^2 - \frac{1}{24}\theta^4$. Therefore, using Lemma 3.5,

$$\|\Phi(x) - \Phi(y)\|_2^2 \geq \frac{1}{m}\sum_{j=1}^m \left((Vz)_j\right)^2 - \frac{1}{12m}\sum_{j=1}^m \left((Vz)_j\right)^4$$

$$= Q(z) - \frac{1}{12}W(z), \quad (8)$$

where we have denoted

$$W(z) := \frac{1}{m}\sum_{j=1}^m \left((Vz)_j\right)^4.$$

We have already controlled the deviation of $Q(z)$ in Equation (7), so we focus on bounding the deviation of $W(z)$. To this end, we use its Wiener chaos decomposition. We briefly review some basics: the $k$th Wiener chaos is the closure of homogeneous degree-$k$ polynomials in a Gaussian vector $Z$. It is spanned by multivariate Hermite polynomials of total degree $k$. An $L^2$-function of $Z$ can be uniquely decomposed as the sum of chaos terms. For more background on the subject, see, e.g., Sullivan (2015); Hairer (2021).

For our purpose, let $h_k(\cdot)$ denote the $k$th "probabilist's" Hermite polynomial. Recall that the 2nd and 4th ones are

$$h_2(t) = t^2 - 1 \quad \text{and} \quad h_4(t) = t^4 - 6t^2 + 3.$$

They satisfy the identity

$$t^4 - 3 = 6h_2(t) + h_4(t). \quad (9)$$

It will be convenient to normalize the entries of $Vz$. Denote

$$Z_j = \frac{1}{\|z\|_2}(Vz)_j \quad \text{and} \quad Y = \frac{1}{m}\sum_{j=1}^m Z_j^4 = \frac{1}{\|z\|_2^4}W(z).$$

By Lemma 3.3, $\forall_j Z_j \sim N(0,1)$. Recalling that the 4th moment of $N(0,1)$ is 3, we have $\mathbb{E}[Y] = 3$. Setting $t = Z_j$ and averaging over $j = 1, \ldots, m$ in Equation (9), we get

$$Y - \mathbb{E}[Y] = \frac{6}{m}\sum_{j=1}^m h_2(Z_j) + \frac{1}{m}\sum_{j=1}^m h_4(Z_j). \quad (10)$$

This is the Wiener chaos decomposition of $Y$, and it has only a 2nd and a 4th chaos term. We denote them by

$$Y_2 := \frac{6}{m}\sum_{j=1}^m h_2(Z_j) \quad \text{and} \quad Y_4 := \frac{1}{m}\sum_{j=1}^m h_4(Z_j).$$

$Y_2$ can be controlled through the same Bernstein analysis from the previous section. Observe that

$$Y_2 = \frac{6}{m}\sum_{j=1}^m \left(Z_j^2 - 1\right) = \frac{6}{\|z\|_2^2}\left(Q(z) - \|z\|_2^2\right),$$

hence by Equation (7),

$$\Pr\left[|Y_2| \leq 6\varepsilon\right] > 1 - \eta. \quad (11)$$

To control $Y_4$ we use a concentration bound for Wiener chaoses, which follows from their hypercontractivity.

**Theorem 3.6.** *Let $X$ be a random variable in the $k$-th Wiener chaos. For all $q > 1$ and $t > e \cdot (\mathbb{E}[|X|^q])^{1/q}$,*

$$\Pr\left[|X| > t\right] < \exp\left(-1 - (q-1)\cdot\left(\frac{t}{e(\mathbb{E}[|X|^q])^{1/q}}\right)^{2/k}\right). \quad (12)$$

*Proof.* The Wiener chaos hypercontractivity theorem is:

**Theorem 3.7** (e.g., Theorem 7.1 in (Hairer, 2021))**.** *Let $X$ be a random variable in the $k$th Wiener chaos. Then for all $p, q \in (1, \infty)$ with $\frac{p-1}{q-1} \geq 1$,*

$$\left(\mathbb{E}[|X|^p]\right)^{1/p} \leq \left(\frac{p-1}{q-1}\right)^{k/2}\left(\mathbb{E}[|X|^q]\right)^{1/q}.$$

Denote $M_q := (\mathbb{E}[|X|^q])^{1/q}$ for brevity. Let $q > 1$ and $t > eM_q$. Choose

$$p = 1 + (q-1)\left(\frac{t}{eM_q}\right)^{2/k}.$$

Note that $q > 1$ implies $q - 1$, and $t > eM_q$ implies $\frac{p-1}{q-1} \geq 1$. Therefore the conditions of Theorem 3.7 are satisfied. Therefore, using Markov's inequality,

$$\Pr\left[|X| > t\right] = \Pr\left[|X|^p > t^p\right] < \frac{M_p^p}{t^p}$$

$$= \left(\frac{M_p}{t}\right)^p \leq \left(\frac{1}{t}\cdot\left(\frac{p-1}{q-1}\right)^{k/2}\cdot M_q\right)^p = e^{-p}.$$

Observing that the right-hand side in Equation (12) is $e^{-p}$, Theorem 3.6 is proven. $\qquad\square$

$Y_4$ is in the $k = 4$ chaos and we choose $q = 2$, obtaining

$$\Pr\left[|Y_4| > t\right] < \exp\left(-O(1) \cdot \sqrt{\frac{t}{\mathbb{E}[(Y_4^2)]^{1/2}}}\right). \quad (13)$$

Hence now we need to bound the second moment of $Y_4$. We use a lemma on chaos correlations, see Appendix A.

**Lemma 3.8.** *Let $X \sim N(0,1)$, $Y \sim N(0,1)$ have correlation $\rho_{XY}$. Then for every $k$,*

$$\mathbb{E}[h_k(X)h_k(Y)] = k!\rho_{XY}^k.$$

*Proof.* The generating function of the probabilist's Hermite polynomials $\{h_n\}$ is

$$\exp(xt - \tfrac{1}{2}t^2) = \sum_{n=0}^{\infty} h_n(x)\frac{t^n}{n!}. \quad (14)$$

From the joint moment generating function of Gaussians,

$$\mathbb{E}[\exp(tX + sY)] = \exp\left(\tfrac{1}{2}t^2 + \tfrac{1}{2}s^2 + ts\rho_{XY}\right),$$

which rearranges to

$$\mathbb{E}\left[\exp\left(tX - \tfrac{1}{2}t^2 + sY - \tfrac{1}{2}s^2\right)\right] = \exp(ts\rho_{XY}).$$

By Equation (14), the left-hand side is expanded as

$$\mathbb{E}\left[\exp\left(tX - \tfrac{1}{2}t^2 + sY - \tfrac{1}{2}s^2\right)\right]$$
$$= \sum_{n=1}^{\infty}\sum_{m=1}^{\infty} \mathbb{E}[h_n(X)h_m(Y)]\frac{t^n s^m}{n!m!}.$$

The right-hand side is expanded as as

$$\exp(ts\rho_{XY}) = \sum_{k=1}^{\infty} \frac{(ts\rho_{XY})^k}{k!}.$$

Considering the coefficient of $(ts)^k$ on both sides, we get

$$\mathbb{E}[h_k(X)h_k(Y)] \cdot \frac{1}{(k!)^2} = \frac{\rho_{XY}^k}{k!},$$

which rearranges to the lemma statement. $\qquad \square$

Let $\Gamma \in \mathbb{R}^{m \times m}$ be the correlation matrix between the $Z_j$s. Recall that

$$Z_j = \frac{1}{\|z\|_2}(Vz)_j = \frac{\sqrt{m}}{\|z\|_2}(HGu)_j.$$

Then,

$$\Gamma_{ij} = \mathbb{E}[Z_i Z_j]$$
$$= \frac{m}{\|z\|_2^2}\mathbb{E}\left[\sum_{i'=1}^{m} H_{ii'}g_{i'}u_{i'}\sum_{j'=1}^{m} H_{jj'}g_{j'}u_{j'}\right]$$
$$= \frac{m}{\|z\|_2^2}\sum_{i'=1}^{m}\sum_{j'=1}^{m} H_{ii'}H_{jj'}u_{i'}u_{j'}\mathbb{E}[g_{i'}g_{j'}]$$
$$= \frac{m}{\|z\|_2^2}\sum_{\ell=1}^{m} H_{i\ell}H_{j\ell}u_\ell^2,$$

or written in matrix form,

$$\Gamma = \frac{m}{\|z\|_2^2} \cdot H\text{diag}\left(u^2\right)H^T. \quad (15)$$

Therefore,

$$\mathbb{E}[Y_4^2] = \frac{1}{m^2}\sum_{i=1}^{m}\sum_{j=1}^{m}\mathbb{E}[h_4(Z_i)h_4(Z_j)]$$
$$= \frac{4!}{m^2}\sum_{i=1}^{m}\sum_{j=1}^{m}\Gamma_{ij}^4 \qquad \text{Lemma 3.8}$$
$$\leq \frac{4!}{m^2}\sum_{i=1}^{m}\sum_{j=1}^{m}\Gamma_{ij}^2 \qquad \forall_{ij}|\Gamma_{ij}| \leq 1$$
$$= \frac{4!}{\|z\|_2^4}\|H\text{diag}\left(u^2\right)H^T\|_F^2 \qquad \text{Equation (15)}$$
$$= \frac{4!}{\|z\|_2^4}\|\text{diag}\left(u^2\right)\|_F^2 \qquad H \text{ is orthogonal}$$
$$= \frac{4!}{\|z\|_2^4}\sum_{j=1}^{m} u_j^4$$
$$\leq \frac{24L_z^2}{\|z\|_2^2} \qquad \text{Equation (6)}$$
$$= \frac{48\log(2m/\eta)}{m}. \qquad \text{Equation (3)}$$

Plugging this in Equation (13) with $t = 1$, Equation (3) and our setting of $m$ (here it suffices that $m \gg \log^4(1/\eta)$),

$$\Pr\left[|Y_4| \leq 1\right] \geq 1 - \eta.$$

Plugging this with Equation (11) in Equation (10) we get $\Pr[|Y| \leq 3 + 6\varepsilon + 1] \geq 1 - 2\eta$. Under this event, recalling that $Y = \frac{1}{\|z\|_2^4}W(z)$, and that in item 2 of Theorem 1.3 we have the premise $\|z\|_2^2 \leq \varepsilon$, we have

$$|W(z)| \leq (4 + 6\varepsilon)\|z\|_2^4 \leq O(\varepsilon)\|z\|_2^2.$$

Finally, taking a last union bound with Equation (7) and plugging into Equation (8), we have with probability $1 - 3\eta$,

$$\|\Phi(x) - \Phi(y)\|_2^2 \geq Q(z) - \tfrac{1}{12}W(z) \geq (1 - O(\varepsilon))\|z\|_2^2,$$

proving item 2 of Theorem 1.3.

### 3.4. Distance Collapse

Lastly, we prove item 3 in Theorem 1.3, that with high probability $\Phi$ does not collapse squared distances to be less than $\Omega(\varepsilon)$. We use a tail bound for Lipschitz functions of Gaussians (see, e.g., Boucheron et al. (2003, Theorem 5.6)) as also used by Le et al. (2013). Applying this tail bound to $f_z$ which was defined in Lemma 3.4, we have

$$\forall\, t > 0, \quad \Pr\left[|f_z(g) - \mathbb{E}[f_z(g)]| > t\right] < 2\exp(-t^2/2L_z^2).$$

It is not difficult to verify that (see Lemma A.3),

$$\mathbb{E}[f_z(g)] = e^{-\|z\|_2^2/2}.$$

We choose

$$t_z = 1 - \tfrac{1}{8}\varepsilon - e^{-\|z\|_2^2/2}.$$

For all $\varepsilon \leq 0.75$ it can be checked that $1 - \tfrac{1}{8}\varepsilon > e^{-\varepsilon/2}$. Also, under item 3 of Theorem 1.3 we have the premise $\|z\|_2^2 > \varepsilon$. Together these ensure that $t_z > 0$. Therefore, we have the bound

$$\Pr\left[f_z(g) < 1 - \varepsilon/8\right] \geq 1 - 2\exp(-t_z^2/2L_z^2). \quad (16)$$

Note that Lemma 3.5 can be written as

$$\|\Phi(x) - \Phi(y)\|_2^2 = 2 - 2f_z(g).$$

Hence, under the event in (16) we have $\|\Phi(x) - \Phi(y)\|_2^2 \geq \tfrac{1}{4}\varepsilon$ as needed. So it remains to verify that the probability on the right-hand side of (16) is at least $1 - O(\eta)$.

We analyze two cases. In the first case, $\varepsilon < \|z\|_2^2 \leq 1$. By a Taylor expansion we have $e^{-\theta} < 1 - \theta + \tfrac{1}{2}\theta^2$, hence

$$e^{-\|z\|_2^2/2} \leq 1 - \frac{1}{2}\|z\|_2^2 + \frac{1}{8}\|z\|_2^4.$$

Since $\|z\|_2^2 \leq 1$, this implies $e^{-\|z\|_2^2/2} \leq 1 - \tfrac{3}{8}\|z\|_2^2$. Hence, $t_z > \tfrac{1}{4}\|z\|_2^2$. With Equation (3), the failure probability in (16) is now at most $\exp(-\widetilde{\Omega}(\|z\|_2^2 \cdot m))$. Since $\|z\|_2^2 > \varepsilon$, it suffices that $m \gg 1/\varepsilon$ for it to be at most $\eta$.

In the complement case, $1 < \|z\|_2^2 \leq \Lambda^2$. From $1 < \|z\|_2^2$ and $\varepsilon < 1/2$ we get $t_z \geq 1 - \tfrac{1}{8} - e^{-1/2} > 1/4$. With Equation (3), the failure probability in (16) is at most

$$\exp(-\widetilde{\Omega}(\|z\|_2^{-2}m)) \leq \exp(-\widetilde{\Omega}(\Lambda^{-2}m)).$$

Thus, it suffices that $m \gg \Lambda^2$ for it to be at most $\eta$. Since our choice of $m$ satisfies the conditions in both cases, item 3 is proven, and the proof of Theorem 1.3 is complete.

## 4. Proof of Theorem 1.2

By Lemma A.1, we may assume w.l.o.g. that $|X| = O(\log(1/\delta)/\varepsilon^2)$ by subsampling it down to that size as preprocessing. Lemma A.1 ensures that with probability $1 - \delta$ this preserves the KDE for all $y \in \mathbb{R}^d$ up to $\pm\varepsilon$.

We will use the following Fastfood concentration result for the Gaussian kernel due to Le et al. (2013, Theorem 11).

**Theorem 4.1** (Le et al. (2013)). *Let $\varepsilon, \eta \in (0, 1)$. Let $\mathbf{k}(x, y)$ be the Gaussian kernel over $\mathbb{R}^m$. There is a randomized map $F : \mathbb{R}^m \to \mathbb{R}^\ell$ with $\ell = \widetilde{O}(\widehat{\Delta}^2/\varepsilon^2)$, computable in time $\widetilde{O}(m + \ell)$, such that for each pair $x, y \in \mathbb{B}^d(\widehat{\Delta})$:*

$$\Pr\left[\left|\mathbf{k}(x, y) - F(x)^T F(y)\right| < \varepsilon\right] > 1 - \eta.$$

Let $s = \sqrt{\varepsilon/(c\log(1/\varepsilon))}$ where $c \geq 1$ is a constant we will choose later. We define the scaled spherical embedding $\Psi(x) := \Phi(sx)/s$, where $\Phi$ is from Theorem 1.3 instantiated with $\Lambda = 2s\Delta \leq \sqrt{\varepsilon}\Delta$. We then define the map $K(x) := F(\Psi(x))$, where $F$ is the map from Theorem 4.1 invoked with diameter $\widehat{\Delta} = 2/s$. In both Theorems 1.3 and 4.1 we set $\eta = \delta/|X|$ to allow for a union bound over the pairs $x, y$ for a fixed query $y$ and all $x \in X$.

Our algorithm maps every point $x \in \mathbb{B}^d(\Delta)$ to the random features $K(x)$. At construction time we compute $K(X) := \frac{1}{|X|}\sum_{x \in X} K(x)$. At query time, we return $K(X)^T K(y)$.

**Query time.** Applying $\Phi$ to a query $y \in \mathbb{R}^d$ takes time $\widetilde{O}(d + \varepsilon\Delta^2 + \varepsilon^{-2})$ by Theorem 1.3. $\Phi$ outputs points on the unit sphere, hence $\Psi$ outputs points on the sphere with radius $2/s$, hence the target dimension in Theorem 4.1 is $\ell = \widetilde{O}(1/(s^2\varepsilon^2)) = \widetilde{O}(\varepsilon^{-3})$. Thus, applying $F$ to $y$ takes time $\widetilde{O}(d + \varepsilon\Delta^2 + \varepsilon^{-3})$, and the inner product $K(X)^T K(y)$ takes time $\widetilde{O}(\varepsilon^{-3})$. The total query time is thus $\widetilde{O}(d + \varepsilon\Delta^2 + \varepsilon^{-3})$.

**Accuracy.** The following lemma draws the connection between our spherical embedding and KDE.

**Lemma 4.2.** *For every fixed pair $x, y \in \mathbb{B}^d(\Delta)$,*

$$\Pr\left[\left|e^{-\|\Psi(x) - \Psi(y)\|_2^2} - e^{-\|x - y\|_2^2}\right| < 2\varepsilon\right] \geq 1 - \eta.$$

*Proof.* We consider two cases. In the first case, $\|x - y\|_2^2 \leq c\log(1/\varepsilon)$. Hence, $\|sx - sy\|_2^2 \leq \varepsilon$. By Theorem 1.3, we have $\|\Phi(sx) - \Phi(sy)\|_2^2 = (1 \pm \varepsilon)\|sx - sy\|_2^2$. with probability $1 - \eta$. Hence, under this event, $\|\Psi(x) - \Psi(y)\|_2^2 = (1 \pm \varepsilon)\|x - y\|_2^2$. It is not hard to verify (see Lemma A.4) that this implies

$$\left|e^{-\|\Psi(x) - \Psi(y)\|_2^2} - e^{-\|x - y\|_2^2}\right| < \varepsilon.$$

In the second case, $\|x - y\|_2^2 > c\log(1/\varepsilon)$. This implies $e^{-\|x - y\|_2^2} \leq \varepsilon^c \leq \varepsilon$. Since $\|x - y\|_2 \leq \Delta$, we also have $\|sx - sy\|_2^2 \leq s^2\Delta^2 \leq \Lambda^2$. Hence, by Theorem 1.3, $\|\Phi(sx) - \Phi(sy)\|_2^2 \geq \Omega(\varepsilon)$ with probability $1 - \eta$. Under this event we have

$$\|\Psi(x) - \Psi(y)\|_2^2 \geq \frac{\Omega(\varepsilon)}{s^2} = \Omega(\varepsilon) \cdot \frac{c\log(1/\varepsilon)}{\varepsilon} \geq \log(1/\varepsilon),$$

where the last inequality holds provided $c$ is chosen as a sufficiently large constant to offset the $\Omega(\cdot)$ in item 3 of Theorem 1.3. This inequality is equivalent to $e^{-\|\Psi(x) - \Psi(y)\|_2^2} < \varepsilon$. Thus,

$$\left|e^{-\|x - y\|_2^2} - e^{-\|\Psi(x) - \Psi(y)\|_2^2}\right|$$
$$\leq e^{-\|x - y\|_2^2} + e^{-\|\Psi(x) - \Psi(y)\|_2^2} \leq 2\varepsilon.$$

Thus, the conclusion of the lemma holds in both cases. $\square$

To prove Theorem 1.2, let $x, y \in \mathbb{B}^d(\Delta)$. By Lemma 4.2 we have $|e^{-\|x-y\|_2^2} - e^{-\|\Psi(x)-\Psi(y)\|_2^2}| \le 2\varepsilon$ with probability $1 - \eta$. By Theorem 4.1, we have $|e^{-\|\Psi(x)-\Psi(y)\|_2^2} - F(\Psi(x))^T F(\Psi(y))| \le \varepsilon$ with probability $1 - \eta$. Together,

$$\Pr\left[\left|e^{-\|x-y\|_2^2} - K(x)^T K(y)\right| \le 3\varepsilon\right] > 1 - 2\eta. \quad (17)$$

Recall we have chosen $\eta$ to allow for a union bound over all pairs $\{(x, y) : x \in X\}$. Averaging the event in (17) over them and rescaling constants yields Theorem 1.2.

## 5. Additional Related Work

Orthogonal Random Features (ORF) (Yu et al., 2016; Choromanski et al., 2017; 2018), Quasi Monte-Carlo features (QMC) (Avron et al., 2016; Huang et al., 2024) and quadrature-based features (Bach, 2017; Munkhoeva et al., 2018) offer extensions and alternatives to RFF for kernel approximation with reduced variance. These lines of work are complementary to the worst-case query time bounds we study and yield the same $O(d/\varepsilon^2)$ bound as RFF in this context. Notably, Yu et al. (2016) proposed Structured ORF (SORF), a triple-iterated RHT method ($\psi(SHD_3HD_2HD_1x)$ in the notation from Section 2), as a heuristic alternative to ORF, and showed it performs well empirically. The triple-RHT heuristic was also proposed in Andoni et al. (2015) for locality sensitive hashing (LSH). Kernel Nyström methods (Rudi et al., 2015; Gittens & Mahoney, 2016; Musco & Musco, 2017) produce data-dependent features that control the error based on properties of the kernel matrix, though not in the worst case.

KDE coresets are a well-studied tool for reducing the data size $|X|$ to a size independent of its original value (Lopez-Paz et al., 2015; Lacoste-Julien et al., 2015; Karnin & Liberty, 2019; Phillips & Tai, 2020; Dwivedi & Mackey, 2024). As mentioned earlier, they are useful as preprocessing in our method to avoid any dependence on $|X|$ in the query time.

KDE data structures with relative error were pioneered by Charikar & Siminelakis (2017) and widely studied since (Siminelakis et al., 2019; Charikar et al., 2020; Backurs et al., 2018; 2019; Kapralov et al., 2026). This stronger error guarantee entails longer query times than those in Table 1. Backurs et al. (2024) proposed hybrid additive-relative error KDE data structures via preprocessing relative error KDEs with FJLT. The same can be done with our spherical embedding, Theorem 1.3, in place of FJLT, for improved hybrid running times.

KDE data structures have also recently been used for approximate kernel matrix-vector multiplication, with applications to efficient Attention in long-context transformer-based deep learning architectures (Choromanski et al., 2021; Backurs et al., 2021; Zandieh et al., 2023; Bakshi et al., 2023; Shah et al., 2025; Carrell et al., 2025; Indyk et al., 2025).

## 6. Discussion, Limitations and Open Questions

We proved a new bound on the time complexity of kernel mean queries, improving in some cases over classical algorithms. Our techniques introduce a new fast spherical embedding theorem and involve a new chaos-based analysis of randomized Hadamard transforms.

Our spherical embedding result Theorem 1.3 may be of independent interest. Bartal et al. (2011)'s embedding it is based on has been used in a range of other applications including local (cardinality-based) dimension reduction, Euclidean snowflake dimension reduction, private near neighbor counting in high dimensions, Euclidean graph spanners, and efficient algorithms for the Earth Mover Distance (Bartal et al., 2011; Gottlieb & Krauthgamer, 2015; Andoni et al., 2023; Andoni & Zhang, 2023). Theorem 1.3 could possibly be used to improve running time bounds in these areas.

The KDE query times in Table 1 paint a rather complex mosaic of techniques and parameter regimes. The main open question is to establish optimal bounds for this fundamental problem. In particular, it is interesting whether other trade-offs between the error $\varepsilon$ and the effective diameter $\Delta_\sigma$ are possible and what is the optimal trade-off between them. Is a $\widetilde{O}(d + 1/\varepsilon^2)$ bound possible that depends on $\Delta_\sigma$ only polylogarithmically, or even not at all?

Another open question is extensions to other radial kernels (i.e., which are a function of the Euclidean distance). Theorem 1.4 extends our Gaussian kernel result to IMQ kernels with essentially the same bound, through a substantive function approximation result due to Beylkin & Monzón (2010). Extensions to other kernels may be handled along similar lines, though they might require specialized work and the adapted bound might degrade according to the specific kernel. This limitation of our method is shared by Fastfood and FJLT+RFF, which also necessitate per-kernel treatment, and they are similarly specialized to radial kernels. RFF on its own is less sensitive to the specific kernel and is not specialized to radial kernels. It yields the same $O(d/\varepsilon^2)$ bound as long as the kernel is positive definite, shift-invariant, and its spectral measure is efficient to sample from.

Our results are primarily theoretical, and we do not include an empirical evaluation. This is partly because structured matrix methods like RHTs are highly sensitive to hardware and implementation, and require specialized care in practice (see, e.g., Andersson & Karppa (2026)), whereas full matrix multiplication is typically already highly optimized and parallelized. We leave this undertaking for future work. Prior work (e.g., Yu et al. (2016)) has reported that heuristic methods for kernel approximation can match or outperform provable methods empirically (see, e.g., their comparison of SORF and Fastfood), and it remains an open problem to bridge the gap between them.

## Acknowledgements

This research was supported by the Israeli Ministry of Innovation, Science & Technology, by Len Blavatnik and the Blavatnik Family foundation, and by an Alon Scholarship.

## Impact Statement

This paper presents work whose goal is to advance the field of Machine Learning. There are many potential societal consequences of our work, none which we feel must be specifically highlighted here.

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

## A. Auxiliary Lemmas for Sections 3 and 4

**Lemma A.1** (Gaussian KDE coreset by random sampling (Lopez-Paz et al., 2015; Phillips & Tai, 2020)). *Let $X \subset \mathbb{R}^d$. Let $X'$ be a uniformly random subsample (with replacement) from $X$ of size $O(\log(1/\delta)/\varepsilon^2)$. Then, for the Gaussian kernel $\mathbf{k}(x, y) = \exp(-\|x - y\|_2^2/\sigma^2)$ with any bandwidth $\sigma > 0$,*

$$\Pr_{X'} \left[ \forall\, y \in \mathbb{R}^d, \quad \left| \frac{1}{|X|} \sum_{x \in X} \mathbf{k}(x, y) - \frac{1}{|X'|} \sum_{x \in X'} \mathbf{k}(x, y) \right| \le \varepsilon \right] > 1 - \delta.$$

This coreset lemma allows us to assume w.l.o.g. that the ground set $X$ for Gaussian KDE has size $O(\log(1/\delta)/\varepsilon^2)$. Note that Lemma A.1 provides a "for all" guarantee: with probability $1 - \delta$, the sampled coreset $X'$ preserves the KDE for all $y \in \mathbb{R}^d$ simultaneously. For our purposes, the following weaker "for each" guarantee suffices:

$$\text{For each fixed } y \in \mathbb{R}^d, \qquad \Pr_{X'} \left[ \left| \frac{1}{|X|} \sum_{x \in X} \mathbf{k}(x, y) - \frac{1}{|X'|} \sum_{x \in X'} \mathbf{k}(x, y) \right| \le \varepsilon \right] > 1 - \delta.$$

This "for each" guarantee is an immediate consequence of Hoeffding's inequality for any kernel $\mathbf{k}$ that takes values in $[0, 1]$.

**Lemma A.2** (Lemma 3.5 restated). $\|\Phi(x) - \Phi(y)\|_2^2 = \frac{2}{m} \sum_{j=1}^{m} \left(1 - \cos\left((Vz)_j\right)\right)$.

*Proof.*

$$\begin{aligned}
\|\Phi(x) - \Phi(y)\|_2^2 &= \frac{1}{m} \sum_{j=1}^{m} \left( (\cos((Vx)_j) - \cos((Vy)_j))^2 + (\sin((Vx)_j) - \sin((Vy)_j))^2 \right) \\
&= \frac{1}{m} \sum_{j=1}^{m} (2 - 2\cos((Vx)_j)\cos((Vy)_j) - 2\sin((Vx)_j)\sin((Vy)_j)) \\
&= \frac{1}{m} \sum_{j=1}^{m} (2 - 2\cos((Vx)_j - (Vy)_j)) \\
&= \frac{2}{m} \sum_{j=1}^{m} (1 - \cos((Vz)_j)),
\end{aligned}$$

where the third equality is by a standard trigonometric sum-product identity, and the fourth equality is since $(Vx)_j - (Vy)_j = (V(x - y))_j = (Vz)_j$. □

**Lemma A.3** (used in Section 3.4). *Let $f_z : \mathbb{R}^m \to \mathbb{R}$ be the map defined in Lemma 3.4:*

$$f_z(g) = \frac{1}{m} \sum_{j=1}^{m} \cos((Vz)_j).$$

*For every fixed diagonal sign matrix $B$ in Equation (1), it holds that*

$$\mathbb{E}[f_z(g)] = \exp(-\|z\|_2^2/2).$$

*Proof.* We recall the standard fact that $\mathbb{E}[\cos(X)] = \exp(-\nu^2/2)$ for $X \sim N(0, \nu^2)$. Thus it suffices to establish that $(Vz)_j \sim N(0, \|z\|_2^2)$ for every $j$. Indeed, $(Vz)_j = \sqrt{m} \sum_{i=1}^{m} H_{ji} u_i g_i$ and $g \sim N(0, I)$, hence $(Vz)_j$ is distributed like $N(0, \nu^2)$ with $\nu^2 = m \sum_{i=1}^{m} (H_{ji} u_i)^2 = \sum_{i=1}^{m} u_i^2 = \|u\|_2^2 = \|z\|_2^2$. We have used the fact that each entry of $H$ is in $\{\pm 1/\sqrt{m}\}$ for the second equality, and Equation (2) for the last equality. □

**Lemma A.4** (used in Section 4). *Let $\varepsilon \in (0, 1 - \frac{1}{e})$ and let $r_1, r_2 \ge 0$ be such that $r_2 = (1 \pm \varepsilon)r_1$. Then $|e^{-r_1} - e^{-r_2}| < \varepsilon$.*

*Proof.* By the mean value theorem there is $\rho \in [\min(r_1, r_2), \max(r_1, r_2)]$ such that $|e^{-r_1} - e^{-r_2}| \le e^{-\rho}|r_1 - r_2|$. The premise $r_2 = (1 \pm \varepsilon)r_1$ is equivalent to $|r_1 - r_2| \le \varepsilon r_1$, and $\rho \ge \min(r_1, r_2)$ implies $\rho \ge (1 - \varepsilon)r_1$, hence

$$e^{-\rho}|r_1 - r_2| \le \varepsilon r_1 e^{-(1-\varepsilon)r_1} \le \varepsilon \cdot \sup_{r \ge 0} r e^{-(1-\varepsilon)r} = \varepsilon \cdot \frac{1}{e(1 - \varepsilon)} \le \varepsilon.$$

□

# B. Proof of Theorem 1.4: Inverse Multi-Quadratic Kernels via Function Approximation

Let $\mathbf{k}(x, y) = 1/(1 + \|x - y\|_2^2)^\beta$ be the $\beta$-IMQ kernel.

We use the following function approximation theorem due to Beylkin & Monzón (2010).

**Theorem B.1** (Beylkin & Monzón (2010)). *Let $\beta > 0$, $\zeta \in (0, 1]$ and $\varepsilon \in (0, 1/e)$. There are $h > 0$ and integers $M, N$ that satisfy:*

$$\forall\, r \in [\zeta, 1] \quad \left| r^{-\beta} - \frac{h}{\Gamma(\beta)} \sum_{\ell=M+1}^{N} e^{\beta h \ell} e^{-e^{h\ell} r} \right| \leq r^{-\beta} \varepsilon,$$

*and furthermore:*

- $h = \Theta(1/(\beta + \log(\varepsilon^{-1})))$.

- $N - M \leq \widetilde{O}((\log(1/\varepsilon) + \log \beta)(\log(1/\zeta) + \beta^{-1} \log(1/\varepsilon) + \log \log(1/\varepsilon)))$.

- $hN \leq h + t^*$ *for* $t^* = \log \beta + \log(1/\zeta) + \log \log(1/\varepsilon) + O(1)$.

Let $x, y \in \mathbb{B}^d(\Delta)$. Let

$$\zeta := \frac{1}{1 + \Delta^2} \qquad \text{and} \qquad r := \frac{1 + \|x - y\|_2^2}{1 + \Delta^2}.$$

Since $\|x - y\|_2 \leq \Delta$ we have $r \in [\zeta, 1]$. Therefore, denoting

$$\lambda_\ell := \zeta \cdot e^{h\ell} \qquad \text{and} \qquad \alpha_\ell := \frac{\zeta^\beta h e^{\beta h \ell - \lambda_\ell}}{\Gamma(\beta)},$$

we have by Theorem B.1,

$$\left| \left( \frac{1}{1 + \|x - y\|_2^2} \right)^\beta - \sum_{\ell=M+1}^{N} \alpha_\ell e^{-\lambda_\ell \cdot \|x - y\|_2^2} \right| < \left( \frac{1}{1 + \|x - y\|_2^2} \right)^\beta \varepsilon \leq \varepsilon. \tag{18}$$

Since this holds in particular for $x = y$, we have

$$\sum_{i=M+1}^{N} \alpha_i \leq 1 + \varepsilon. \tag{19}$$

Furthermore, by the sum formula for a geometric series,

$$\sum_{\ell=M+1}^{N} \lambda_\ell = \zeta \sum_{\ell=M+1}^{N} e^{h\ell} = \zeta \cdot \frac{e^{h(M+1)}(e^{h(N-M)} - 1)}{e^h - 1} \leq \zeta \cdot \frac{e^{hN+h}}{e^h - 1} \leq \zeta \cdot \frac{e^{t^*+2h}}{h}, \tag{20}$$

where $t^*$ is from Theorem B.1 and we have used that $e^h - 1 \geq h$. By Theorem B.1 we have

$$e^{2h} \leq O(1) \quad \text{and} \quad \frac{1}{h} \leq O(\beta + \log(1/\varepsilon)) \quad \text{and} \quad e^{t^*} \leq O\left( \frac{\beta \log(1/\varepsilon)}{\zeta} \right),$$

and plugging these in Equation (20), we get

$$\sum_{\ell=M+1}^{N} \lambda_\ell \leq O\left( \beta \log(1/\varepsilon) \cdot (\beta + \log(1/\varepsilon)) \right) = \widetilde{O}(\beta^2). \tag{21}$$

Now suppose we have a dataset $x_1, \ldots, x_n \in \mathbb{B}^d(\Delta)$. We construct a Gaussian KDE data structure $\mathcal{G}_\ell$ from Theorem 1.2 for every $\ell = M + 1, \ldots, N$, with distances scaled by up by $\sqrt{\lambda_\ell}$ (or in other words, with bandwidth $1/\sqrt{\lambda_\ell}$). In each $\mathcal{G}_\ell$ we set the prescribed error to $\varepsilon$ and the failure probability to $\delta' = \delta/(N - M) \geq \widetilde{O}(\delta/(1 + \log \beta))$ to allow for a union bound.

Upon receiving a query $y \in \mathbb{B}^d(\Delta)$ we query all of the $\mathcal{G}_\ell$s. The total query time is

$$\sum_{\ell=M+1}^{N} \widetilde{O}\left(d + \varepsilon^{-3} + \varepsilon \Delta^2 \lambda_\ell\right) = \widetilde{O}\left((N-M)\left(d + \varepsilon^{-3}\right) + \varepsilon \Delta^2 \sum_{\ell=M+1}^{N} \lambda_\ell\right)$$

$$= \widetilde{O}\left((1 + \log(\beta))\left(d + \varepsilon^{-3}\right) + \varepsilon \Delta^2 \beta^2\right), \tag{22}$$

having used Equation (21) and the bound on $N - M$ from Theorem B.1.

For accuracy, let $E_\ell$ denote the Gaussian KDE estimate returned by $\mathcal{G}_\ell$. Then we have,

$$\left| \frac{1}{n} \sum_{i=1}^{n} \left(\frac{1}{1 + \|x_i - y\|_2^2}\right)^\beta - \sum_{\ell=M+1}^{N} \alpha_\ell E_\ell \right|$$

$$= \left| \frac{1}{n} \sum_{i=1}^{n} \left(\frac{1}{1 + \|x_i - y\|_2^2}\right)^\beta - \frac{1}{n} \sum_{i=1}^{n} \sum_{\ell=M+1}^{N} \alpha_\ell e^{-\lambda_\ell \|x_i - y\|_2^2} + \frac{1}{n} \sum_{i=1}^{n} \sum_{\ell=M+1}^{N} \alpha_\ell e^{-\lambda_\ell \|x_i - y\|_2^2} - \sum_{\ell=M+1}^{N} \alpha_\ell E_\ell \right|$$

$$\leq \frac{1}{n} \sum_{i=1}^{n} \left| \left(\frac{1}{1 + \|x_i - y\|_2^2}\right)^\beta - \sum_{\ell=M+1}^{N} \alpha_\ell e^{-\lambda_\ell \|x_i - y\|_2^2} \right| + \sum_{\ell=M+1}^{N} \alpha_\ell \left| \frac{1}{n} \sum_{i=1}^{n} e^{-\lambda_\ell \|x_i - y\|_2^2} - E_\ell \right|$$

$$\leq \varepsilon + \sum_{\ell=M+1}^{N} \alpha_\ell \left| \frac{1}{n} \sum_{i=1}^{n} e^{-\lambda_\ell \|x_i - y\|_2^2} - E_\ell \right| \qquad \text{by Equation (18).} \tag{23}$$

To bound the second term in (23), by Theorem 1.2 with a union bound over all $\ell$, we have with probability $1 - \widetilde{O}(\delta)$ that

$$\forall \, \ell, \qquad \left| \frac{1}{n} \sum_{i=1}^{n} e^{-\lambda_\ell \|x_i - y\|_2^2} - E_\ell \right| \leq \varepsilon.$$

Therefore, with probability $1 - \widetilde{O}(\delta)$,

$$\left| \frac{1}{n} \sum_{i=1}^{n} \left(\frac{1}{1 + \|x_i - y\|_2^2}\right)^\beta - \sum_{\ell=M+1}^{N} \alpha_\ell E_\ell \right| \leq \varepsilon + \sum_{\ell=M+1}^{N} \alpha_\ell \left| \frac{1}{n} \sum_{i=1}^{n} e^{-\lambda_\ell \|x_i - y\|_2^2} - E_\ell \right| \qquad \text{by Equation (23)}$$

$$\leq \varepsilon + \sum_{\ell=M+1}^{N} \alpha_\ell \varepsilon$$

$$\leq \varepsilon + (1 + \varepsilon)\varepsilon = O(\varepsilon) \qquad \text{by Equation (19).}$$

Theorem 1.4 follows by rescaling hidden constants and polylog factors.

## C. Proof of Theorem 1.5: Differentially Private KDE

Differential privacy (Dwork et al., 2006) has become a central area in algorithms and machine learning. DP KDE has been studied in (Hall et al., 2013; Wang et al., 2016; Alda & Rubinstein, 2017; Coleman & Shrivastava, 2021; Wagner et al., 2023; Backurs et al., 2024; Wagner, 2025). We refer to Dwork & Roth (2014) for background and basic definitions of DP.

To clarify notation, let us emphasize that we focus on pure DP with a single DP parameter $\varepsilon_{\mathrm{DP}}$.[2] The parameter $\delta$ in this section is the failure probability of attaining the requisite approximation guarantee (similarly to its role in Theorem 1.2), and not a secondary privacy parameter for approximate DP.

We focus on DP KDE in the function release model recently studied in Coleman & Shrivastava (2021); Wagner et al. (2023); Backurs et al. (2024). It is a variant of Definition 1.1 in which the output of the construction stage is required to be $\varepsilon_{\mathrm{DP}}$.

---

[2]The extension to approximate DP is straightforward (by replacing the Laplace mechanism with the Gaussian mechanism, see Dwork & Roth (2014)) and is omitted.

**Definition C.1.** Let $\mathbf{k} : \mathbb{R}^d \times \mathbb{R}^d \to \mathbb{R}$ be a kernel. An $(\varepsilon, \delta, \varepsilon_{\mathrm{DP}})$-DP KDE function release mechanism $\mathcal{M}$ takes a finite set $X \subset \mathbb{R}^d$ as input and outputs a function description $\widetilde{E}(\cdot)$ which is $\varepsilon_{\mathrm{DP}}$-DP w.r.t. $X$ and such that for each fixed query $y \in \mathbb{R}^d$,

$$\Pr\left[\left|\frac{1}{|X|}\sum_{x \in X} \mathbf{k}(x, y) - \widetilde{E}(y)\right| < \varepsilon\right] > 1 - \delta,$$

where the probability is over the internal randomness of $\mathcal{M}$.

Wagner et al. (2023) proposed a framework for DP KDE and applied it to RFF. Backurs et al. (2024) use the same framework to adapt their FJLT+RFF approach to DP KDE. The framework introduces the condition that $|X| \geq O(\log(1/\delta)/(\varepsilon_{\mathrm{DP}}\varepsilon^2))$.[3] However, the framework relies on the random features being i.i.d. This is satisfied for RFF and FJLT+RFF owing to their use of a full random Gaussian matrix, but it does not cover Fastfood nor our methods, which have probabilistically dependent features as a result of the final RHT they apply.

We prove Theorem 1.5 by showing how to adapt our method from Theorem 1.2 to DP KDE with essentially the same condition as RFF and FJLT+RFF. The same proof can be applied to Fastfood as well. Our approach is to apply an FJLT transform on our method's output feature vector. Indeed, this means yet another RHT (followed by a sampling and scaling matrix), obtaining a feature map of the form

$$f_{\mathrm{ours-DP}}(x) = SHD_5 \cdot \psi(HD_4 HD_3 \cdot s^{-1}\psi(HD_2 HD_1(sx))),$$

where $SHD_5$ is an FJLT transform into $\widetilde{O}(1/\varepsilon^2)$ coordinates, and the rest of the matrices are as in Theorem 1.2. We then truncate the entries and apply DP noise.

The motivation behind this added FJLT is that the KDE methods discussed in this paper ultimately use an inner product estimator over approximate linear features to approximate the KDE. On the one hand, FJLT has the JL dimensionality reduction property, which means it preserves that inner product up to a bounded error with high probability. On the other hand, FJLT has a flattening property (Lemma Theorem 3.2) which controls the magnitude of entries of the vectors it outputs, rendering it useful for applying privacy-preserving noise. We now prove this formally.

*Proof of Theorem 1.5.* Let $K : \mathbb{R}^d \to \mathbb{R}^m$ be the feature map from Theorem 1.2. It is defined in Section 4 and its accuracy guarantee is stated in Equation (17). Recall that $m = \widetilde{O}(d + \varepsilon\Delta_\sigma^2 + 1/\varepsilon^3)$. Let $\delta' = \delta/|X|$. For convenience we denote $v_x = K(x)$ for every $x \in \mathbb{R}^d$.

Let $\ell = O(\log(m/\delta')\log(1/\delta')/\varepsilon^2)$. Let $A \in \mathbb{R}^{\ell \times m}$ be the FJLT matrix,

$$A = \sqrt{\frac{m}{\ell}} SHD,$$

where $D \in \mathbb{R}^{m \times m}$ is diagonal with i.i.d. uniformly random signs on the diagonal, and $S \in \mathbb{R}^{\ell \times m}$ has i.i.d. rows sampled at random from the standard basis in $\mathbb{R}^m$. By Ailon & Chazelle (2009), this matrix satisfies the JL property, and in particular, for each fixed pair of unit-norm vectors $v, u \in \mathbb{R}^m$ it satisfies

$$\Pr_{S,D}\left[\left|v^T u - (Av)^T(Au)\right| \leq \varepsilon\right] > 1 - \delta'. \tag{24}$$

We also denote

$$L := \sqrt{\frac{2\log(4m/\delta')}{\ell}} = O(1) \cdot \frac{\varepsilon}{\sqrt{\log(1/\delta')}},$$

where the right-hand side equality is by plugging our setting of $\ell$.

Our DP-KDE mechanism works as follows:

1. Sample an FJLT matrix $A$ as above.

---

[3]The intuition for the necessity of the condition is that that dataset $X$ needs to be large enough to enable accurate KDE approximations while preserving the privacy of each element in the dataset.

2. For every $x \in X$,

    (a) Compute $v_x = K(x)$ and $a_x = Av_x$.

    (b) Truncate each entry of $a_x$ into the interval $[-L, L]$. Denote the resulting vector $\widehat{a}_x$.

3. Compute the vector $M_X := \frac{1}{|X|} \sum_{x \in X} \widehat{a}_x$.

4. Add to each entry of $M_X$ noise sampled from $\mathrm{Laplace}(2\ell L / (\varepsilon_{\mathrm{DP}}|X|))$. Call the resulting vector $\widetilde{M}_X$ and release it together with $K$ and $A$.

5. Given a query point $y \in \mathbb{R}^d$, the KDE estimate is $\widetilde{E}(y) := \widetilde{M}_X^T A K(y)$.

Since each $\widehat{a}_x$ is $\ell$-dimensional and satisfies $\|\widehat{a}_x\|_\infty \leq L$, its $\ell_1$-sensitivity is $2\ell L$. Therefore, the $\ell_1$-sensitivity of the mean vector $M_X$ is $2\ell L / |X|$. It follows from the Laplace DP mechanism that $\widetilde{M}_X$ is $\varepsilon_{\mathrm{DP}}$-DP. We refer to Dwork & Roth (2014) for the standard definition of the $\ell_1$-sensitivity and the guarantee of the Laplace DP mechanism. Since $K$ and $A$ are random maps sampled obliviously of any data, they can be released without impacting DP. Also observe that computing $\widehat{a}_x$ from $v_x$ takes time $O(m \log m)$ (dominated by the Walsh-Hadamard transform in $A = SHD$), and therefore, the time to compute $M_X$ is dominated by computing $v_x = K(x)$ per $x \in X$. This is the running time from Theorem 1.2.

We now prove that the mechanism satisfies the accuracy guarantee from Definition Theorem C.1. Fix $x \in \mathbb{R}^d$. By the flattening property of FJLT (the rectangular form of Lemma 3.2, see Ailon & Chazelle (2009)), we have

$$\Pr_D \left[ \|a_x\|_\infty \leq L \|v_x\|_2 \right] \geq 1 - \delta'. \tag{25}$$

Recall that Fastfood (and hence our map $K$) outputs vectors with unit norm (since the entries it outputs are normalized sine/cosine pairs). Hence, $\|v_x\|_2 = 1$. Therefore, the event in Equation (25) implies that $\|a_x\|_\infty \leq L$, in which case the truncation from $a_x$ to $\widehat{a}_x$ has no effect. In summary, for each fixed $x \in \mathbb{R}^d$ we have

$$\Pr_D \left[ \widehat{a}_x = a_x \right] \geq 1 - \delta'.$$

By a union bound, this occurs for all $x \in X$ simultaneously with probability $1 - \delta'|X| = 1 - \delta$. We condition on this event and fix $y \in \mathbb{R}^d$. Then the DP-KDE estimate is

$$\widetilde{E}(y) := \widetilde{M}_X^T A K(y) = M_X^T a_y + N^T a_y, \tag{26}$$

where $N \in \mathbb{R}^\ell$ has i.i.d. entries sampled from $\mathrm{Laplace}(b)$ with $b = 2\ell L / (\varepsilon_{\mathrm{DP}}|X|)$. Since $N^T a_y = \sum_{j=1}^\ell a_{y,j} N_j$ is a subexponential random variable, we have by a Bernstein concentration bound,

$$\Pr_N \left[ |N^T a_y| \leq O(1) \cdot b \cdot \left( \|a_y\|_2 \sqrt{\log(2/\delta')} + \|a_y\|_\infty \log(2/\delta') \right) \right] > 1 - \delta'.$$

As discussed above, by the flattening property of FJLT we have with probability $1 - O(\delta')$ that $\|a_y\|_\infty \leq L < 1$, and by the JL property of FJLT we have with probability $1 - O(\delta')$ that $\|a_y\|_2^2 \leq (1 + \varepsilon)\|v_y\|_2^2 = 1 + \varepsilon < 2$. Therefore, with probability $1 - O(\delta')$ we have $|N^T a_y| \leq \widetilde{O}(b)$.

We proceed to the term $M_X^T a_y$ in Equation (26). First, recall that by Theorem 1.2, we have with probability $1 - \delta$,

$$\left| \frac{1}{|X|} \sum_{x \in X} \mathbf{k}(x, y) - \frac{1}{|X|} \sum_{x \in X} v_x^T v_y \right| < \varepsilon.$$

We also condition on the event in Equation (24) simultaneously for all pairs $\{(v_x, v_y) : x \in X\}$. By a union bound, this occurs with probability $1 - \delta'|X| = 1 - \delta$. Under this event,

$$\left| \frac{1}{|X|} \sum_{x \in X} v_x^T v_y - \frac{1}{|X|} \sum_{x \in X} (Av_x)^T (Av_y) \right| \leq \frac{1}{|X|} \sum_{x \in X} \left| v_x^T v_y - (Av_x)^T (Av_y) \right| \leq \frac{1}{|X|} \sum_{x \in X} \varepsilon = \varepsilon.$$

For the DP KDE estimate $M_X^T a_y$, we have,

$$M_X^T a_y = \frac{1}{|X|} \sum_{x \in X} \widehat{a}_x^T a_y = \frac{1}{|X|} \sum_{x \in X} a_x^T a_y = \frac{1}{|X|} \sum_{x \in X} (Av_x)^T (Av_y),$$

recalling that we are conditioning on $\widehat{a}_x = a_x$ for all $x \in X$. Putting everything together, with probability $1 - O(\delta)$, we have obtained

$$\left| \frac{1}{|X|} \sum_{x \in X} \mathbf{k}(x, y) - \widetilde{E}(y) \right|$$

$$\leq \left| \frac{1}{|X|} \sum_{x \in X} \mathbf{k}(x, y) - M_X^T a_y \right| + |N^T a_y|$$

$$\leq \left| \frac{1}{|X|} \sum_{x \in X} \mathbf{k}(x, y) - \frac{1}{|X|} \sum_{x \in X} (Av_x)^T (Av_y) \right| + |N^T a_y|$$

$$\leq \left| \frac{1}{|X|} \sum_{x \in X} \mathbf{k}(x, y) - \frac{1}{|X|} \sum_{x \in X} v_x^T v_y \right| + \left| \frac{1}{|X|} \sum_{x \in X} v_x^T v_y - \frac{1}{|X|} \sum_{x \in X} (Av_x)^T (Av_y) \right| + |N^T a_y|$$

$$\leq \widetilde{O}\left( \varepsilon + \frac{\ell L}{\varepsilon_{\mathrm{DP}} |X|} \right).$$

Note that $\ell L = \widetilde{O}(1/\varepsilon)$, and therefore the total error is $\widetilde{O}\left( \varepsilon + \frac{1}{\varepsilon \cdot \varepsilon_{\mathrm{DP}} |X|} \right)$. Under the premise in Theorem 1.5 that $|X| \geq \widetilde{O}(1/(\varepsilon^2 \varepsilon_{\mathrm{DP}}))$, the total error is $\widetilde{O}(\varepsilon)$, and we can rescale it by a polylog factor to become $\varepsilon$ by absorbing the polylog in the $\widetilde{O}(\cdot)$ notation without changing the statement of the theorem. We similarly scale the failure probability by $\widetilde{O}(1)$ to $\delta$. This concludes the proof of Theorem 1.5. $\qquad\square$

The proof for adapting Fastfood to DP is identical except that the map $K$ is replaced by the map $F$ from Theorem 4.1.

## D. Self-Contained Algorithm Description

To enhance clarity, this appendix contains a self-contained algorithmic description of the data structure from Theorem 1.2, given next.

**Setup and notation:**

- Let $m = \widetilde{O}(d + \varepsilon \Delta_\sigma^2 + 1/\varepsilon^{-3})$, with a sufficiently large hidden $\mathrm{polylog}(d, \Delta_\sigma, \varepsilon^{-1}, \delta^{-1})$ factor and such that $m$ is a power of 2.

- Let $s = \Theta(\sqrt{\varepsilon / \log(1/\varepsilon)})$ with a sufficiently small hidden constant.

- For an integer $\ell > 0$, let $\psi_\ell : \mathbb{R}^\ell \to \mathbb{R}^{2\ell}$ denote the map from $\ell$ coordinates to their normalized sine/cosine pairs:

$$\forall j = 1, \dots, \ell, \qquad \psi_\ell(x)_{2j-2} = \frac{1}{\sqrt{\ell}} \cos(x_j) \quad \text{and} \quad \psi_\ell(x)_{2j-1} = \frac{1}{\sqrt{\ell}} \sin(x_j).$$

- For an integer $\ell > 0$ which is a power of 2, let $h_\ell : \mathbb{R}^\ell \to \mathbb{R}^\ell$ denote the normalized Walsh-Hadamard transform on $\ell$-dimensional vectors.

- For every $x \in \mathbb{R}^d$, denote by $\bar{x} \in \mathbb{R}^m$ its zero-padding up to $m$ dimensions.

**Construction:**

- Sample random diagonal matrices $B_1 \in \mathbb{R}^{m \times m}$ and $B_2 \in \mathbb{R}^{2m \times 2m}$ with uniform i.i.d. signs (Rademachers) on the diagonal, and random diagonal matrices $G_1 \in \mathbb{R}^{m \times m}$ and $G_2 \in \mathbb{R}^{2m \times 2m}$ with i.i.d. $N(0, 1)$ on the diagonal.

- Define the map $K : \mathbb{R}^d \to \mathbb{R}^{4m}$ as

$$K(x) := \psi_{2m}(h_{2m}(G_2 h_{2m}(B_2 \cdot s^{-1} \psi_m(h_m(G_1 h_m(B_1 \cdot s\bar{x})))))).$$

- Compute and store $\bar{K}(X) := \frac{1}{|X|} \sum_{x \in X} K(x)$.

**Query:** Compute and return $\bar{K}(X)^T K(y)$.

*Remark.* There is a small difference in the output dimension between the algorithm description here and its implicit description in the proof of Theorem 1.2 in the main text, though it does not impact the running time (asymptotically). Here, we define $K(x)$ as having output dimension $4m$ to simplify the algorithm's presentation. In contrast, in Section 4 we invoke Theorem 4.1 as it appears in Le et al. (2013), which entails reducing the output dimension of the final Walsh-Hadamard transform to $\widetilde{O}(\widehat{\Delta}/\varepsilon^2)$ (where in our case $\widehat{\Delta} = 1/\sqrt{\varepsilon}$) by randomly sampling coordinates. The subsampling step is already "baked into" Theorem 4.1, so we keep it there to simplify the proof. In either case, the query time is dominated by evaluating $K(y)$ in time $\widetilde{O}(m)$, so the asymptotic bound in Theorem 1.2 remains the same.

