# OpenReview forum: "New Bounds for Kernel Sums via Fast Spherical Embeddings"
_ICML.cc/2026/Conference — ICML 2026 regular_

### Official Review · Reviewer_YNYA · 2026-02-19

**Soundness:** 4
**Presentation:** 3
**Significance:** 4
**Originality:** 3
**Overall Recommendation:** 5
**Confidence:** 3

**Summary:**

This paper studies improved query-time bounds for kernel density estimation (KDE), focusing on the Gaussian kernel in high-dimensional settings.
The authors investigate randomized feature-based kernel approximation and refine the trade-offs between dimension $d$, approximation error $\varepsilon$, and effective diameter $\Delta_\sigma$.
The main contribution is a new KDE data structure which improves upon prior approaches (RFF, FJLT+RFF, and Fastfood) in regimes characterized by small $\varepsilon$ and intermediate effective diameter.

The technical core is a new fast spherical embedding theorem, serving as an FJLT-style analog of Bartal et al. (2011).
The embedding preserves small distances up to $(1 \pm \varepsilon)$ distortion, prevents moderate distances from collapsing, and reduces effective diameter in a way that enables improved Fastfood-based approximation.

**Compliance With Llm Reviewing Policy:**

Affirmed.

**Final Justification:**

This paper seeks to analyze the concept of improving kernel density estimation query time through structured embeddings, and the authors attempt to study a general domain of high-dimensional kernel approximation with careful theoretical guarantees.

Overall, I recommend acceptance. The paper is technically strong, with solid theoretical soundness and a clear improvement over prior bounds in meaningful parameter regimes. The introduction of the fast spherical embedding is original and potentially of independent interest, which strengthens the paper’s significance.

The rebuttal addressed my main concerns.

**Key Questions For Authors:**

Can you provide concrete numerical examples illustrating regimes where your method clearly dominates Fastfood and FJLT+RFF?

Does the spherical embedding stage introduce noticeable overhead in practical implementations?

Is the $1/\varepsilon^3$ dependence inherent to the approach, or could it potentially be improved?

**Limitations:**

Yes.

**Strengths And Weaknesses:**

Strengths

The main result provides an improved query-time bound in well-defined parameter regimes.
The comparison table clearly positions the contribution relative to RFF, FJLT+RFF, and Fastfood.

The Wiener chaos-based treatment of fourth-order terms addresses contraction effects that standard Lipschitz concentration bounds would not capture.

Weaknesses

The gains hold primarily in intermediate-diameter and small-error regimes.
In extreme regimes, existing methods may remain competitive or be superior.

---

> ### Author Rebuttal · Authors · 2026-03-27
>
> Thank you for your review and support!
>
> > Can you provide concrete numerical examples illustrating regimes where your method clearly dominates Fastfood and FJLT+RFF?
>
> Please see our response to 7irT which contains an illustration of parameter regimes on real KDE datasets. It includes numerical examples where each of the bounds in Table 1 is better than the other, including several examples where ours dominates:
> * ALOI: $d=128$, $\Delta\_\sigma\leq 46$, $\varepsilon=0.01,0.05,0.1$
> * SVHN: $d=3072$, $\Delta\_\sigma\leq 466.2$, $\varepsilon=0.01,0.05,0.1$
> * MNIST: $d=784$, $\Delta\_\sigma\leq 641.9$, $\varepsilon=0.01,0.05$
>
> > Does the spherical embedding stage introduce noticeable overhead in practical implementations?
>
> In principle, our spherical embedding (Theorem 1.3) entails only RHTs which are practical to implement. The overhead it introduces is two RHTs on top of the two built into Fastfood, for a total of four RHTs. Other iterated RHT methods have been reported as efficient to implement, including SORF (Yu et al., 2016) and FALCONN (Andoni et al., 2015), each entailing three RHTs.
>
> The asymptotic analysis indicates that the cost of the extra RHTs in our spherical embedding should be offset by the smaller output dimension of the overall pipeline, as long as we are in the intermediate effective diameter regime. However, we cannot make definitive claims on practicality as our work is theoretical and we did not produce an implementation.
>
> > Is the $1/\varepsilon^3$ dependence inherent to the approach, or could it potentially be improved?
>
> In our proof, the $1/\varepsilon^3$ term appears because the (effective) diameter becomes $\widehat\Delta = \widetilde O(1/\sqrt{\epsilon})$ after the spherical embedding step, and the subsequent Fastfood invocation (Theorem 4.1) has a $\widehat\Delta^2/\varepsilon^2$ term in the output dimension. In this sense, the dependence is built into our current pipeline: improving it while keeping the exact pipeline intact would mean improving either the critical distance threshold in Theorem 1.3 (meaning the $\sqrt\varepsilon$ threshold where the non-contraction guarantee in item 2 ends and the non-collapse guarantee in item 3 begins) or the output dimension in Fastfood’s tail bound, Theorem 4.1. Both seem difficult and perhaps mathematically impossible.
>
> That said, we do not believe the $1/\varepsilon^3$ term is inherent to the underlying Gaussian KDE problem. In Section 6 we pose improving this dependence as an open question and conjecture that the optimal bound may be $\widetilde O(d+1/\varepsilon^2)$ (possibly with a $\operatorname{polylog}(\Delta_\sigma)$ factor hidden). Achieving that likely requires new ideas beyond our current approach.

---

> > ### Author Rebuttal · Reviewer_YNYA · 2026-04-01
> >
> > My concerns have been adequately addressed.

---

### Official Review · Reviewer_7irT · 2026-03-12

**Soundness:** 3
**Presentation:** 3
**Significance:** 2
**Originality:** 3
**Overall Recommendation:** 4
**Confidence:** 3

**Summary:**

The paper addresses the problem of estimating kernel means for a given set X and a point q. The idea is to compute a sketch of the input vectors in X in a preprocessing step and then to perform a simple operation with the sketch and the query. The problem has been widely studied, and this paper proposes an improvement when the diameter of X is of "intermediate size".
The techniques leverage an improvement of a theorem initially proposed in [Bartal et al., SODA 2011], where the authors now utilize the Hadamard transformation and some results from chaos theory.

**Compliance With Llm Reviewing Policy:**

Affirmed.

**Final Justification:**

I thank the authors for the comments. The rebuttal clarify some concerns and I've slightly increased my scores.

**Key Questions For Authors:**

* Could you stress more the significance of your contribution? how relevant is the range of parameters where you derive your results?

**Limitations:**

yes

**Strengths And Weaknesses:**

Soundness
I haven't checked the results in detail; however, the results are technically sound. The authors build on solid previous works with reasonable assumptions. The provided bounds work on specific parameter ranges of dimension, diameter, and approximations: interestingly, on the extremes of these ranges, the proposed results align with previous works.

Presentation
The technical parts are quite complicated and difficult to follow due to the nature of the proposed methods. Nevertheless, the authors provide an overview of the main techniques, helping readers to understand the main ideas. Furthermore, the paper nicely presents previous works and compares the proposed results with previous ones.

Significance
This is the main weak part of the paper: the contribution improves a quite limited range of the parameters. The authors clearly state this limitation and compare it with previous works. The results improve the state-of-the-art and are technically interesting, but are not particularly surprising.

Originality
The paper builds on different sets of known techniques, like a theorem from Bartal et al. and Hadarman treansform. The results are non-trivial and interesting. It would be useful to see if the proposed theorem 1.3 can be applied in other problems.


Some minor comments
- Theorem 1.3: the sphere is S^{2m -1} however m is alway expressed in asymptotic notation. Why not just use S^{m}?
- Line 200: "The matrices \Sigma, \Pi are not necessary". Why?
- Plot at page 18: it would be useful to move it in the main paper and comment it to better understand parameter ranges.

---

> ### Author Rebuttal · Authors · 2026-03-27
>
> Thank you for your review and appreciation of the technical merit and presentation of our work.
>
> > Could you stress more the significance of your contribution?
>
> Theorem 1.2 makes progress on a central open theoretical problem for high-dimensional Gaussian KDE: determining the optimal query-time tradeoff among $d$, $\varepsilon$, and $\Delta_\sigma$. Prior work established several incomparable frontier points, but it was unclear whether that frontier was complete. Theorem 1.2 shows it is not: there is a provably distinct regime with a strictly better tradeoff, and reaching it requires genuinely new algorithmic ingredients. The extensions to IMQ kernels and to DP KDE further indicate that the method has broader implications.
>
> Technically, our proof contributes a new algorithmic tool for KDE - a fast spherical embedding strong enough for KDE preprocessing. It turns Bartal et al.’s embedding into an algorithmically useful primitive for Gaussian KDE, which it was not before, since the original dense Gaussian embedding has the same $O(d/\varepsilon^2)$ evaluation cost as RFF.
>
> > how relevant is the range of parameters where you derive your results?
>
> To illustrate concrete parameter regimes, we instantiate the unsuppressed main terms from Table 1 (the expressions in the $\widetilde O(\cdot)$) on six public benchmark datasets from Siminelakis et al. (ICML 2019). This is not an implementation of the methods but an instantiation of the theoretical asymptotic bounds. We use the bandwidth settings from Siminelakis et al. and $\varepsilon=0.01,0.05,0.1$. We estimate $\widetilde{\Delta}\_{\sigma}\in[\Delta_{\sigma},2\Delta\_{\sigma}]$ over a random sample of 100 centers. We report the ordering of bounds (smaller is better), writing $A\sim B$ when the winner and runner-up are within a factor of $5$, and otherwise we order the methods using ”<”.
>
> The rows are sorted by $\widetilde{\Delta}_\sigma$. As it grows, the advantage shifts from diameter-sensitive methods (Fastfood and ours) to diameter-free methods (RFF and RFF+FJLT). The intermediate-diameter regime occurs in a substantial portion of the datasets, and our method’s bound is the smallest in many of the listed settings.
>
> | Dataset | $d$ | $\widetilde{\Delta}\_\sigma$ | $\varepsilon=.01$ | $\varepsilon=.05$ | $\varepsilon=.1$ |
> | :--- | ---: | ---: | :--- | :--- | :--- |
> | GloVe | 100 | 4.9 | **Fastfood** $\sim$ RFF < Ours < RFF+FJLT | **Ours** $\sim$ Fastfood < RFF < RFF+FJLT | **Ours** $\sim$ Fastfood < RFF < RFF+FJLT |
> | ALOI | 128 | 46.0 | **Ours** $\sim$ RFF < Fastfood < RFF+FJLT | **Ours** < RFF < RFF+FJLT < Fastfood | **Ours** < RFF+FJLT < RFF < Fastfood |
> | SVHN | 3072 | 466.2 | **Ours** < RFF < RFF+FJLT < Fastfood | **Ours** < RFF+FJLT < RFF < Fastfood | **RFF+FJLT** $\sim$ Ours < RFF < Fastfood |
> | MNIST | 784 | 641.9 | **Ours** < RFF < RFF+FJLT < Fastfood | **Ours** < RFF+FJLT < RFF < Fastfood | **RFF+FJLT** $\sim$ Ours < RFF < Fastfood |
> | Covtype | 54 | 4304.0 | **RFF** $\sim$ Ours < RFF+FJLT < Fastfood | **RFF** < RFF+FJLT < Ours < Fastfood | **RFF** $\sim$ RFF+FJLT < Ours < Fastfood |
> | MSD | 90 | 26440.2 | **RFF** < Ours < RFF+FJLT < Fastfood | **RFF** $\sim$ RFF+FJLT < Ours < Fastfood | **RFF** $\sim$ RFF+FJLT < Ours < Fastfood |
>
>
> > Theorem 1.3: the sphere is S^{2m -1} however m is always expressed in asymptotic notation. Why not just use S^{m}?
>
> We wrote $S^{2m-1}$ to be consistent with the setting $d=m$ in Section 3.1, as then the cos/sin construction outputs $2m$-dimensional points on the sphere $S^{2m-1}$. However, you are right that the theorem itself can be stated with just $S^m$, we will revise it for clarity.
>
> > "The matrices \Sigma, \Pi are not necessary". Why?
>
> $\Sigma$ (denoted $S$ in Fastfood): Le et al. prove their Gaussian kernel results both with and without $\Sigma$ (e.g., Corollary 10 in their arxiv version: https://arxiv.org/pdf/1408.3060 ). $\Sigma$ is a row scaling matrix that decorrelates row lengths (without it all rows have equal lengths) and is useful for extending Fastfood to other radial kernels (their Section 4.4). In our case, row lengths play no role, and the extension to other kernels requires significant work beyond row scaling (e.g., Theorem 1.4), so $\Sigma$ is not useful in our proofs.
>
> $\Pi$: its role in Le et al. is to mix interactions between the rows of the first and the second RHTs in Fastfood. They need it for their variance bound (Theorem 9) since it makes a statement about each individual output row in Fastfood separately. They do not need it for their concentration bound (Theorem 11, which we invoke as Theorem 4.1) since it argues about an aggregate of all output rows. They call this out in their footnote 3 (page 21 on arxiv). Our Theorem 1.3 also argues about an aggregate of all output coordinates, hence $\Pi$ is similarly unnecessary for its proof.
>
> > Plot at page 18: it would be useful to move it in the main paper and comment it to better understand parameter ranges.
>
> We agree, we will revise this.

---

> > ### Author Rebuttal · Reviewer_7irT · 2026-04-03
> >
> > I thank the authors for the comments. The rebuttal clarify some concerns and I've slightly increased my scores.

---

### Official Review · Reviewer_3LTt · 2026-03-15

**Soundness:** 4
**Presentation:** 4
**Significance:** 3
**Originality:** 3
**Overall Recommendation:** 5
**Confidence:** 4

**Summary:**

This paper proposes new querying bounds for the problem of estimating the kernel mean $\frac{1}{\lvert X \rvert}\sum_{x\in X} \boldsymbol{k}(x,y)$ of a query $y$ in a finite dataset $X$ up to additive error $\epsilon$. This is a well studied problem with existing bounds of $O(d/\epsilon^2)$, $\tilde{O}(d+1/\epsilon^4)$, and $\tilde{O}(d+\Delta^2/\epsilon^2)$. This paper aims to improve the bound at the same time, and proposes that $\tilde{O}(d+\epsilon\Delta^2+1/\epsilon^3)$ is achievable.

This paper does not invent completely new approaches or analysis, however, it smartly combines the existing techniques that yields an improved querying bound and so a new fast spherical embedding technique. The strategy is to treat "small" and "large" distances differently. While the preference is given to the "smaller" distance, the embedding only needs to prevent the "large" distance from corrupting. This idea was explored in the prior work of Bartal et al. (2011), while direct application still yield to undesired bound $O(d/\epsilon^2)$ because of the full Gaussian matrix calculation. The authors then combine the techniques from the Fastfood approach of Le et al. (2013) with shorter running time comparing to the full Gaussian transformation. The Fastfood transformation is used at two different places, i.e. in Theorem 4.1 and Theorem 1.3, respectively. Combining the distance treatment and faster transformation techniques, this paper obtains a favorable querying time bound of $\tilde{O}(d+\epsilon\Delta^2+1/\epsilon^3)$.

The work focuses on Gaussian kernel. However, the result can also be extended to the inverse multi-quadratic kernels and to differential privacy kernel density functions.

**Compliance With Llm Reviewing Policy:**

Affirmed.

**Key Questions For Authors:**

The authors have mentioned that some techniques could be of independent interest, i.e. the fast spherical embeddings. Section 6 has answered its possible application in improving the running time in related areas. I wonder if the Wiener chaos decomposition could also find broader applications.

**Limitations:**

yes

**Strengths And Weaknesses:**

This paper studies an important problem of kernel density estimation (KDE) under the $(\epsilon,\delta)$-KDE data structure framework. While being well studied, the author proposes a strictly more favorable bound comparing to the state-of-the-art, in terms of dimension $d$, accuracy parameter $\epsilon$, and spherical radius $\Delta$.

While the idea of separating the small distances from the large distances was already explored in prior work of Bartal et al. (2011) in a different application of dimensionality reduction, the application of this idea in this paper requires mastery of all related technique. As a result, I found the new fast spherical embedding mechanism quite novel and could be of independent interest.

The paper is very well written with all technical challenges well explained and brought upfront, making its contribution very clear. Though I had some doubt while reading the paper, all my doubts have been answered in later sections, which is also a sign of being well-written. The proofs look sound.

---

> ### Author Rebuttal · Authors · 2026-03-27
>
> Thank you for your review and support!
>
> > I wonder if the Wiener chaos decomposition could also find broader applications.
>
> Thank you for the thoughtful question. We do believe the Wiener-chaos step may be useful beyond the present proof. It seems especially applicable when a Taylor bound for an RHT-based nonlinear map introduces quartic (or higher order) terms that need to be controlled. In our case, this entered through the lower bound
> $1-\cos(\theta)\geq \frac12\theta^2-\frac1{24}\theta^4$,
> where the cosine nonlinearity comes from RFF/BRS. This produced both a quadratic and a quartic term, and the latter was controlled through the fourth-chaos argument.
>
> A concrete candidate application is SORF. In the notation of Section 2, it is the triple-RHT map $\psi(SHD_3HD_2HD_1x)$, with the same cosine/sine nonlinearity as RFF. Yu et al. (2016) introduced it as an empirically effective heuristic alternative to RFF/ORF and posed proving rigorous guarantees for it as an open question. We are hopeful that our technique could be useful for making progress on that problem. Another candidate is the triple-RHT heuristic proposed by Andoni et al. (2015) as a replacement for the dense Gaussian matrix in their cross-polytope LSH for the angular distance. In their case too the triple-RHT heuristic lacks a rigorous proof.
>
> We did not emphasize this more strongly in the paper because we do not yet have concrete progress to report. While the applicability of our technique seems plausible in these settings, establishing such results remains uncertain and subject to future work.

---

> > ### Author Rebuttal · Reviewer_3LTt · 2026-04-03
> >
> > My concerns have been addressed.

---

### Official Review · Reviewer_Jgra · 2026-03-16

**Soundness:** 2
**Presentation:** 1
**Significance:** 2
**Originality:** 2
**Overall Recommendation:** 2
**Confidence:** 4

**Summary:**

**Summary**

Given a point set $X \subset \mathbb{R}^d$ and a query point $q \in \mathbb{R}^d$, the paper studies the problem of estimating the Gaussian kernel mean
$KDE(X,q) = \frac{1}{|X|}\sum_{x\in X} \mathbf{k}(x,q)$,
where the kernel function is $\mathbf{k}(x,q) = \exp(-\|x-q\|_2^2/\sigma^2)$ and $\sigma$ denotes the bandwidth parameter. Let $\Delta$ be an upper bound on the diameter of the set $X \cup \{q\}$. The goal is to design a randomized estimator that, with probability at least $1-\delta$, approximates the Gaussian kernel mean within additive error $\epsilon$. In other words, the objective is to construct an $(\epsilon,\delta)$-KDE data structure capable of answering arbitrary queries efficiently.

Three previously known query-time bounds for this problem are $O(d/\epsilon^2)$, $\tilde{O}(d+\epsilon^{-4})$, and $\tilde{O}(d+\epsilon^{-2}\Delta^2)$. The paper proposes a slight improvement of the third bound, obtaining a query time of $\tilde{O}(d + \epsilon \Delta^2 + \epsilon^{-3})$.

**Main Techniques**

The main technical ingredient is a fast spherical embedding theorem inspired by the work of Bartal, Recht, and Schulman on dimensionality reduction beyond the Johnson--Lindenstrauss bound. The approach is largely based on their technique. The main modification is the use of two successive Randomized Hadamard Transforms (RHTs) in the embedding $\Phi(Wx)$, replacing the matrix $W$ with a composition of two RHTs.

In addition, the analysis uses a case distinction for small and large distances based on results from Bartal et al., which is applied effectively in the paper.

**Assessment of the Contribution**

The improvement in query time appears limited. As stated by the authors, ``We prove a new time complexity bound for KDE queries, improving in some regimes over cornerstone methods like RFF, FJLT and Fastfood.'' However, the actual improvement is modest.

Compared with the second known bound $\tilde{O}(d+\epsilon^{-4})$, the new result reduces a factor of $\epsilon^{-1}$ but introduces an additive term $\epsilon\Delta^2$. Compared with the third bound $\tilde{O}(d+\epsilon^{-2}\Delta^2)$, the dependence on $\Delta^2$ improves from $\epsilon^{-2}\Delta^2$ to $\epsilon\Delta^2$, but this comes at the cost of an additional $\epsilon^{-3}$ term. Overall, it is not clear in which parameter regimes this bound provides a meaningful practical improvement over existing results.

Furthermore, most of the techniques are based heavily on the work of Bartal et al., and the novelty of the technical contribution appears limited.

**Comments on the Writing**

The paper would benefit from significant revisions in terms of clarity and presentation. For example, the abstract mentions three bounds but does not explicitly state what these bounds are. The abstract should clearly list them. The statement ``The best known bounds for the Gaussian kernel are \ldots'' is ambiguous and should clarify whether these refer to different estimators, different algorithms, or simply query-time guarantees.

Similarly, the phrase "We prove the new bound ..." should specify precisely whether this refers to query complexity or another quantity. Another issue in the abstract is the sentence "At the center of our proof is a new fast spherical embedding theorem in the sense introduced ...". This paragraph is unclear. If the contribution is a new embedding technique, it would be clearer to simply state that the proof relies on a new fast embedding method rather than framing it primarily through the work of Bartal, Recht, and Schulman. In addition, the paragraph consists of a very long sentence that should be broken into multiple sentences for clarity.

The statement ``This fast embedding theorem may be of independent interest'' is also questionable. If the authors believe this is the case, they should provide concrete examples or potential applications that justify this claim.

**Comments on the Introduction**

The introduction briefly states that estimating empirical kernels has machine learning applications, but it does not provide concrete examples or citations. A single sentence mentioning ML applications without supporting references gives the impression that this connection was added superficially. The authors should expand this discussion and cite relevant applications.

Definition 1.1 is mostly clear. However, the sentence ``Our main interest is in minimizing the query time'' should appear outside the formal definition. It would also be helpful to explain why query time is the main bottleneck: computing the kernel between the query and every point in $X$ requires linear time per query.

If the paper focuses specifically on Gaussian KDE, as suggested later in the introduction, this should be stated clearly in the abstract.

**Comments on the Results**

The notion of query time on the first page is not clearly explained. For instance, when evaluating the kernel between two points, is this assumed to take $O(1)$ time? The computational model should be clarified.

The paper also states that ``The bounds are incomparable and depend on the interplay between the parameters. The first two bounds entail no limitation on the diameter.'' However, the third bound appears less appealing in practice. In many applications the diameter $\Delta$ can be quite large, and the quadratic dependence on $\Delta$ significantly weakens the bound. It is therefore unclear why this regime should be considered attractive merely to reduce the dependence on $1/\epsilon$.

Similarly, the improvement in Theorem 1.2 does not seem particularly significant. The result improves one factor of $1/\epsilon$ relative to the second bound and reduces the term $\Delta^2/\epsilon^2$ to $\epsilon\Delta^2$, but this does not appear substantial enough for a high-level venue such as ICML.

The justification comparing the result with Fastfood is also not convincing. In many regimes, the bound $O(d + 1/\epsilon^4)$ appears more appealing than the proposed bound or the Fastfood approach.

**Additional Comments**

The phrasing ``Like Fastfood, and unlike RFF and RFF+FJLT'' is somewhat informal for a research paper and should be revised.

In Section 2, the notation $d_\epsilon = \Theta(1/\epsilon^2)$ may be confusing because it resembles the dimension $d$. It would be better to use a different symbol to avoid confusion.

**Compliance With Llm Reviewing Policy:**

Affirmed.

**Key Questions For Authors:**

**Key Questions for the Authors**

One important question concerns the practical regime in which the proposed bound improves over previous results. The new query time is $\tilde{O}(d + \epsilon\Delta^2 + \epsilon^{-3})$. It would be helpful if the authors could clearly describe the parameter regimes of $\epsilon$, $\Delta$, and $d$ where this bound is strictly better than the previously known bounds $O(d/\epsilon^2)$, $\tilde{O}(d+\epsilon^{-4})$, and $\tilde{O}(d+\epsilon^{-2}\Delta^2)$. Without such a comparison it is difficult to understand the practical significance of the result.

A related question concerns the dependence on the diameter $\Delta$. In many practical datasets the diameter can be quite large. Could the authors clarify whether there are natural scenarios or applications where the dependence $\epsilon\Delta^2$ leads to an advantage over previous approaches?

The technical contribution relies heavily on the framework of Bartal, Recht, and Schulman. Could the authors elaborate more clearly on what the main conceptual novelty is beyond replacing the embedding matrix $W$ with two Randomized Hadamard Transforms? In particular, which parts of the analysis or construction are fundamentally new?

Another question concerns the fast spherical embedding theorem that is claimed to be of independent interest. Could the authors provide examples of other problems or applications where this embedding result could be useful?

Finally, the abstract and introduction would benefit from clearer explanations of the main results. For example, the abstract refers to the “best known bounds” without explicitly stating them. Could the authors revise this part to clearly summarize the existing bounds and the exact improvement achieved in this work?

**Limitations:**

One limitation of the proposed approach is the dependence of the query time on the diameter parameter $\Delta$. In many practical datasets the diameter of the point set can be quite large, and the term $\epsilon \Delta^2$ may dominate the running time. As a result, the theoretical improvement over previous bounds may not translate into a practical advantage in realistic settings.

Another limitation is that the improvement over prior results is relatively modest. While the dependence on $\Delta^2$ improves compared with the bound $\tilde{O}(d+\epsilon^{-2}\Delta^2)$, this comes at the cost of introducing an additional $\epsilon^{-3}$ term. It is therefore unclear in which parameter regimes the proposed bound provides a clear improvement over existing approaches such as the $\tilde{O}(d+\epsilon^{-4})$ bound.

A further limitation is that the techniques used in the paper rely heavily on existing dimensionality reduction frameworks, particularly the embedding approach of Bartal, Recht, and Schulman. The main modification is the use of two Randomized Hadamard Transforms in the embedding construction. While technically sound, this does not appear to introduce a substantially new conceptual framework.

Finally, the paper does not provide empirical evaluation or experimental validation. Such experiments could help illustrate whether the proposed theoretical improvements lead to measurable gains in practice.

**Strengths And Weaknesses:**

**Strengths**

The paper studies the important problem of estimating the Gaussian kernel mean for arbitrary query points. This is a well-known problem with applications in machine learning, particularly in kernel methods and density estimation. The work attempts to improve the query time of randomized estimators for $(\epsilon,\delta)$-KDE data structures.

The main technical ingredient is a fast spherical embedding theorem inspired by prior work on dimensionality reduction. The approach uses Randomized Hadamard Transforms to construct the embedding efficiently, which is a technically sound direction and builds on established techniques in the literature.

Another positive aspect is that the paper attempts to analyze different parameter regimes and compares the obtained bounds with several well-known approaches such as RFF, FJLT, and Fastfood.

**Weaknesses**

The main weakness of the paper is that the improvement over existing bounds appears rather limited. The proposed query time $\tilde{O}(d + \epsilon\Delta^2 + \epsilon^{-3})$ only slightly improves the dependence on $\Delta^2$ compared with $\tilde{O}(d+\epsilon^{-2}\Delta^2)$, but introduces an additional $\epsilon^{-3}$ term. It is therefore unclear in which parameter regimes this bound leads to a meaningful improvement over previous results.

Another concern is that the techniques used in the paper are heavily based on previous work, particularly the embedding framework of Bartal, Recht, and Schulman. The main modification is the use of two Randomized Hadamard Transforms, which does not appear to introduce a substantially new conceptual idea.

The practical relevance of the bound that depends on $\Delta$ is also unclear. In many realistic datasets the diameter $\Delta$ can be very large, and therefore the quadratic dependence on $\Delta$ may dominate the running time. As a result, the improvement obtained by reducing the dependence on $1/\epsilon$ may not lead to practical benefits.

Finally, the presentation of the paper requires improvement. Several statements in the abstract and introduction are ambiguous, the contributions are not stated clearly, and some parts of the text are written in an informal style that is not suitable for a research paper. More precise explanations and clearer motivation would improve the readability of the paper.

---

> ### Author Rebuttal · Authors · 2026-03-27
>
> Thank you for your review and feedback.
>
> > the abstract refers to the “best known bounds” without explicitly stating them. Could the authors revise this part to clearly summarize the existing bounds and the exact improvement achieved in this work?
>
> > It would be helpful if the authors could clearly describe the parameter regimes of $\epsilon$, $\Delta$ and $d$ where this bound is strictly better than the previously known bounds
>
> To answer directly, the abstract already states those bounds in line 15, and the parameter regimes where each bound is best are listed in the rightmost column of Table 1. If you mean concrete numerical instantiations, we provide this in our response to reviewer 7irT (see below). If we have otherwise misunderstood the question, we are happy to follow up in the second round of response.
>
> > Could the authors clarify whether there are natural scenarios or applications where the dependence $\epsilon \Delta^2$ leads to an advantage over previous approaches?
>
> Our response to 7irT includes an instantiation of the bounds from Table 1 on six public benchmark KDE datasets. They realize a range of regimes: from tightly bounded effective diameter (GloVe), through intermediately bounded effective diameter (ALOI, SVHN, MNIST), to large effective diameter (Covertype, MSD). Our bound is best in 9 of the 18 reported $(\text{dataset},\varepsilon)$ settings.
>
> > The technical contribution relies heavily on the framework of Bartal, Recht, and Schulman. Could the authors elaborate more clearly on what the main conceptual novelty is beyond replacing the embedding matrix $W$ with two Randomized Hadamard Transforms? In particular, which parts of the analysis or construction are fundamentally new?
>
> We do not invoke Bartal et al.'s theorem as is nor a small modification of it. Rather, we use the same spherical embedding definition, but prove a new fast analog from scratch with a different running time. This distinction is essential for KDE: Bartal et al.'s original embedding has the same $O(d/\varepsilon^2)$ evaluation cost as RFF, so it cannot improve KDE query time, while our theorem does.
>
> Analytically, replacing a fully i.i.d. Gaussian matrix by an RHT-based structured transform is not a cosmetic change but a substantial technical step. The difficulty is that one replaces an unstructured matrix with independent entries by a highly dependent transform driven by only $O(d)$ independent random variables (instead of $O(d/\varepsilon^2)$), so the standard concentration tools for dense Gaussian matrices no longer apply directly. In our proof, the genuinely new step is to establish the non-contraction guarantee for this structured spherical embedding, which is where our new chaos-based analysis enters.
>
> In summary, the fundamentally new ingredients in our analysis are (i) the proof of the structured RHT-based spherical embedding theorem itself, and (ii) the fourth-chaos analysis that makes the proof go through.
>
> > Could the authors provide examples of other problems or applications where this embedding result could be useful?
>
> Section 6 provides a list of results in the literature where Bartal et al. (2011)’s embedding is used as a subroutine: local dimension reduction (Bartal et al.’s original application), snowflake dimension reduction (Gottlieb & Krauthgamer, 2015), differentially private near neighbor counting (Andoni et al., 2023), Euclidean graph spanners, and Earth-Mover Distance algorithms (both from Andoni & Zhang, 2023).
>
> Our Theorem 1.3 provides has the same embedding guarantees except that the non-collapse property holds only for distances up to $\Lambda$, with a running time of $\widetilde O(d+1/\varepsilon^2+\Lambda^2)$ versus the original $O(d/\varepsilon^2)$. Consequently, in all the above applications, our theorem can be used instead of Bartal et al.’s and improve the embedding time for inputs with a sufficiently bounded diameter $\Lambda \ll \sqrt{d}/\varepsilon$.
>
> > when evaluating the kernel between two points, is this assumed to take $O(1)$ time? The computational model should be clarified
>
> We use the standard RAM model with unit-cost arithmetic operations without any custom assumptions. Evaluating the Gaussian kernel between two points takes time $O(d)$, dominated by a Euclidean distance computation in $d$ dimensions.

---

> > ### Author Rebuttal · Reviewer_Jgra · 2026-04-03
> >
> > You have not responded to the first two questions (beginning with “One important question concerns the practical regime…”), and I do not see any improvement in how these issues are addressed.
> >
> > To reiterate, one key question concerns the practical regime in which the proposed bound improves over previous results. It would be helpful if the authors could clearly specify the parameter ranges under which the new query time is strictly better than the known bounds. Without such a comparison, it is difficult to assess the practical significance of the result.
> >
> > A related concern is the dependence on the diameter. In many practical datasets, the diameter can be quite large. Could the authors clarify whether there are natural scenarios or applications in which this dependence actually leads to an advantage over prior approaches?
> >
> > As it stands, I have sufficient expertise in this area, and I do not see a significant improvement here.

---

> > > ### Author Response · Authors · 2026-04-04
> > >
> > > Thank you for your continued engagement in the discussion.
> > >
> > > > You have not responded to the first two questions
> > >
> > > These two points were addressed in our prior rebuttal. For ease of reference and to remove as much ambiguity as possible, we restate both answers here.
> > >
> > > > clearly specify the parameter ranges under which the new query time is strictly better than the known bounds
> > >
> > > The parameter range is: $\varepsilon^{-0.5} \lesssim \Delta_\sigma \lesssim \min(\sqrt d\varepsilon^{-1.5}, \varepsilon^{-2.5})$. This appears in our paper in Table 1 on page 2, under "regime where best".
> > >
> > > Interpretation: this is an intermediate effective diameter regime. The lower bound says $\Delta_\sigma$ should not be too small, since then Fastfood is better. The upper bound says $\Delta_\sigma$ should not be too large, since then either RFF or FJLT+RFF is better.
> > >
> > > > Could the authors clarify whether there are natural scenarios or applications in which this dependence actually leads to an advantage over prior approaches
> > >
> > > Such natural scenarios are shown in our response to reviewer 7irT. In it, we instantiated the four bounds on six public KDE datasets spanning small, intermediate, and large effective diameter regimes. Our bound is smallest in half of the reported settings, especially on the datasets ALOI, SVHN, and MNIST which fall in the intermediate effective diameter bracket. Thus this regime arises in natural KDE scenarios.
> > >
> > > > I do not see a significant improvement here
> > >
> > > We summarize the significance of work:
> > > * A new provable trade-off regime for Gaussian KDE query time, making progress on a foundational well-studied problem.
> > > * A new spherical embedding theorem which is useful for KDE, unlike the prior dense Gaussian version whose $O(d/\varepsilon^2)$ evaluation cost is as slow as RFF.
> > > * A new fourth-chaos argument for analyzing structured RHT-based maps.
> > > * Applications to IMQ kernels and to differentially private KDE.
> > >
> > > We believe this constitutes a substantive conceptual and technical advance.

---

### Decision · Program_Chairs · 2026-04-30

**Decision:**

Accept (regular)

**Comment:**

The paper studies faster algorithm for estimating the kernel density at a given query point. Using a new spherical embedding, the paper obtains improved query time for an intermediate regime of diameter compared with the accuracy parameter. All reviewers generally agree that the limited range of parameters where the new method is applicable is the main weakness of the paper. However, several reviewers appreciate the theoretical technique behind the new embedding using chaos analysis. The new algorithm is similar to previous work except it replaces a full gaussian matrix with randomized Hadamard transforms. This is widely used in heuristics but proving correctness is typically challenging. The reviewers' interest in the paper is primarily theoretical so I recommend weak accept.